# ENTROPY-DRIVEN SCANNING OPTIMIZATION FOR NEAR REAL-TIME EARTH OBSERVATION

## ABSTRACT

Earth observation aims to collect geospatial information using remote sensing satellites. However, traditional systems often require days or even weeks to achieve full-region coverage. In this paper, we present the first entropy-based formulation of satellite scanning optimization, designed to enable near real-time Earth observation with large-scale Low Earth Orbit (LEO) constellations. Unlike conventional coverage plans that follow rigid orbital patterns, our approach directly maximizes spatial entropy over imaging point distributions, promoting diversity and fairness in spatiotemporal coverage. This principled objective prevents redundant observations, ensures balanced regional attention, and provides smooth transitions between successive scan plans. To operationalize the framework, we introduce a differentiable solver that maps optimized imaging points into physically executable camera angles, and an efficient satellite-to-task assignment module that minimizes slewing effort through a hybrid of the Hungarian algorithm and nearest-neighbor heuristics. Experimental results demonstrate that our framework achieves full-region coverage within minutes and delivers up to 10× faster scanning compared to conventional orbit-based strategies.

## 1 INTRODUCTION

Low Earth Orbit (LEO) satellite constellations have drastically enhanced global Earth observation capabilities. Unlike legacy monolithic satellites with limited revisit rates, modern constellations—such as Starlink and OneWeb—comprise thousands of agile, coordinated platforms. These infrastructures unlock real-time applications in environmental monitoring Aragon et al. (2018), disaster relief Barmpoutis et al. (2020), precision agriculture Kong et al. (2019), and large-scale infrastructure sensing Chen et al. (2020).

Despite advances in hardware, observation strategies remain outdated. Many systems rely on static or nadir-pointing scanning schedules Nag et al. (2018); Pearl et al. (0), leading to redundant coverage in some regions and long gaps elsewhere. These inefficiencies arise from a lack of coordinated decision-making: satellites act myopically, failing to adapt to global context or long-term sensing goals. Achieving efficient and fair observation at scale requires optimizing how satellites distribute their attention—where and when to observe—across space and time. As shown in Figure 1, the Starlink constellation consists of thousands of satellites continuously orbiting the Earth. By strategically planning the imaging actions of these satellites, it becomes feasible to coordinate their collective sensing capabilities for near real-time Earth observation at global scale.

We tackle this challenge by proposing a distributional formulation of satellite scanning. Instead of treating the task as a series of discrete assignments, we view it as the evolution of a spatial probability distribution over imaging points. At each time step, the system selects a continuous set of imaging targets by optimizing the distribution itself. This enables powerful tools from variational analysis and optimal transport to be applied in this dynamic coordination setting.

Our key innovation is to introduce an entropy-based formulation for satellite scanning optimization. By treating the distribution of imaging points as a probabilistic field and directly maximizing spatial entropy, our framework promotes diverse and balanced coverage rather than allowing observations to collapse into redundant regions. This principled objective yields smooth transitions between successive scanning plans, enforces fairness across regions of interest, and fundamentally departs

from prior heuristics or event-driven scheduling approaches that lack a global information-theoretic perspective.

To operationalize this model, we (i) optimize over a continuous imaging point distribution, (ii) match imaging tasks to satellites with minimal slew cost using a hybrid matching strategy, and (iii) reverse the optimal imaging locations into satellite-specific control angles via a differentiable geometric model. This full-stack pipeline turns the global planning problem into a differentiable and efficiently solvable flow-based optimization.

We evaluate our method using real Starlink TLE data across wildfire-prone regions in California, Colorado, and Texas. Compared to traditional methods, our approach

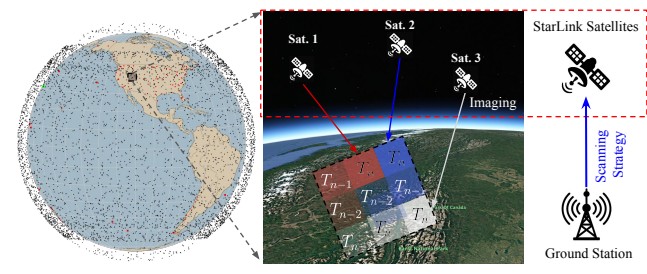

Figure 1: Illustration of utilizing Starlink satellites to execute Earth observation missions. (a) The locations of Starlink satellites sta (2025). The black dots present satellites. (b) An example of three satellites proactively scanning the target area from $T_{n-2}$ to $T_n$. (c) Scanning strategies as signals in ground-space communication: The scanning strategies are computed in ground stations and sent to satellites.

achieves significantly faster coverage, lower revisit gaps, and completes full-region scanning within minutes, offering up to a $10\times$ speedup.

Our contributions can be summarized as follows:

- We formulate satellite imaging as a continuous optimization problem over soft spatial distributions, allowing for scalable and data-driven coordination across large constellations.
- We propose a hybrid objective that balances spatial entropy maximization with temporal consistency, with provable optimality and convergence.
- Our method scales from regional to continental levels, consistently improving coverage speed and scan quality in large-scale simulations.

**Paper Organization.** The remaining paper is organized as follows. In Section 2, we survey relevant research. In Section 3, we explain our proposed methodology. In Section 4, we present experimental verification and result comparisons. In Section 5, we conclude this paper.

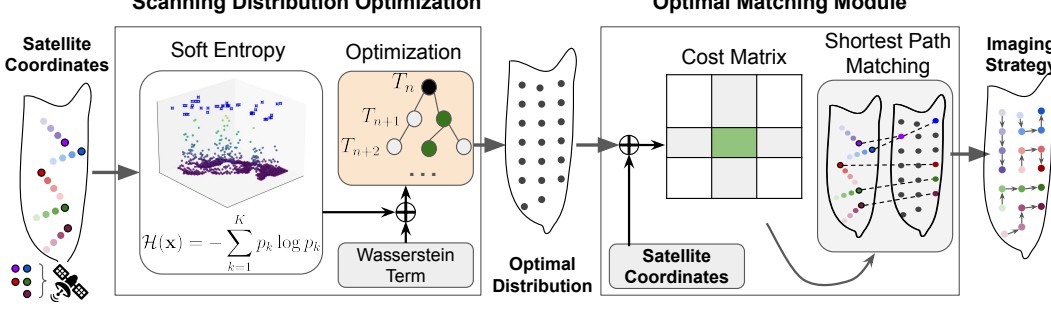

Figure 2: Method Overview. Our method consists of two modules: (i) Scanning Distribution Optimization and (ii) Optimal Matching Module.

## 2 METHOD

In this section, we provide a comprehensive overview of our proposed framework. The framework is illustrated in Figure 2. First, We formulate satellite observation scheduling as a continuous optimization problem over soft assignment distributions, where each candidate imaging action is assigned a probability of execution (Sec. 2.1). Second, we propose scanning distribution optimization to get the optimal distribution of imaging points (Sec. 2.2). Next, we demonstrate the process of the optimal matching module 2.3. Finally, we introduce calculation of controlling angles as practical scanning strategies 2.4.

## 2.1 PROBLEM FORMULATION

We consider a constellation of $N$ LEO satellites $\mathcal{S} = \{s_1, s_2, \ldots, s_N\}$ tasked with imaging a geographic region $\mathcal{A} \subset \mathbb{R}^2$ over a time horizon $[0, T]$. Each satellite follows a predictable orbit and is equipped with a downward-facing sensor with a known field of view (FoV), projected onto the Earth's surface as a rectangular footprint of fixed dimensions: width $W_{imaging}$ and height $H_{imaging}$.

**Imaging Point** An *imaging point* is defined as the geospatial projection of a satellite's optical axis onto the Earth's surface at a given time. For satellite $s_i$ at time $t$, this point is denoted by $I_{i,t} = (\varphi_{i,t}, \lambda_{i,t}) \in \mathbb{R}^2$, where $\varphi_{i,t}$ and $\lambda_{i,t}$ are the latitude and longitude of the point being imaged. The precise location of $p_{i,t}$ depends on the satellite's current geolocation $(\varphi_{i,t}^{sat}, \lambda_{i,t}^{sat})$, altitude $h_{i,t}$, and camera slewing configuration (pitch $\theta_{i,t}^{pitch}$, roll $\theta_{i,t}^{roll}$ as shown in Figure 3, and azimuth angle $\alpha_{i,t}$). The imaging point is computed via:

$$\varphi_{i,t} = \varphi_{i,t}^{sat} + \frac{1}{111000}\left(h_{i,t}\tan\theta_{i,t}^{pitch}\sin\alpha_{i,t} - \frac{h_{i,t}}{\cos\theta_{i,t}^{pitch}}\tan\theta_{i,t}^{roll}\cos\alpha_{i,t}\right), \quad (1)$$

$$\lambda_{i,t} = \lambda_{i,t}^{sat} + \frac{1}{111000 \cdot \cos\varphi_{i,t}}\left(h_{i,t}\tan\theta_{i,t}^{pitch}\cos\alpha_{i,t} - \frac{h_{i,t}}{\cos\theta_{i,t}^{pitch}}\tan\theta_{i,t}^{roll}\sin\alpha_{i,t}\right). \quad (2)$$

**Imaging Point Distribution** Given a satellite's pitch and roll adjustment limits, the space of feasible imaging points forms a bounded region on the Earth's surface, denoted by $\mathcal{I}_{i,t} \subset \mathbb{R}^2$. Specifically, the pitch and roll angles are typically constrained to physical ranges such as $\theta_{i,t}^{pitch} \in [-\theta_p^{max}, \theta_p^{max}]$ and $\theta_{i,t}^{roll} \in [-\theta_r^{max}, \theta_r^{max}]$. These constraints determine the imaging footprint swath:

$$\mathcal{I}_{i,t} = \left\{ I_{i,t}(\theta_{i,t}^{pitch}, \theta_{i,t}^{roll}, \alpha_{i,t}) \mid |\theta_{i,t}^{pitch}| \leq \theta_p^{max}, |\theta_{i,t}^{roll}| \leq \theta_r^{max}, \alpha_{i,t} \in [0, 2\pi] \right\}. \quad (3)$$

The optimization of imaging point selection thus operates over a constrained feasible set $\mathcal{I}_{i,t}$ for each satellite, allowing the scheduler to determine the best combination of slewing angles to maximize spatiotemporal coverage or entropy objectives.

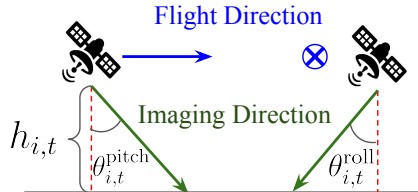

Figure 3: Illustration of adjusting angles of FOV. **Left:** $\theta_{i,t}^{pitch}$ represents the along-track off-nadir angle. **Right:** $\theta_{i,t}^{roll}$ is the cross-track off-nadir angle.

**Soft Grid Probability.** To evaluate how each imaging point influences the local area, we discretize $\mathcal{A}$ into $K$ uniform square grid cells $\{g_1, g_2, \ldots, g_K\}$, each with side length $\Delta_g$ (e.g., 10 km). These cells form the grid set $\mathcal{T}_i$. To simplify notation and focus on the spatial distribution, we omit the time index $t$ of an imaging point $I_{i,t}$ in the subsequent derivation. Each cell $g_k$ receives observation hits based on selected tasks. Based on the grid cells, we can calculate the soft grid probability $p_j$ of a specific area with $n$ imaging points and $K$ small grid cells:

$$p_j = \frac{s_j}{\sum_{k=1}^{K} s_k}, \quad \sum_{j=1}^{K} p_j = 1, \quad (4)$$

where,

$$s_j = \sum_{i=1}^{n} \exp\left(-\frac{d_{ji}}{2\sigma^2}\right), \quad (5)$$

and,

$$d_{ji} = \|\mathbf{g}_j - \mathbf{I}_i\|^2, \quad j = 1, \ldots, K, \ i = 1, \ldots, n \quad (6)$$

**Spatial Diversity Objective.** To promote region-wide, angle-diverse observation, we utilize our defined soft grid probability $p_k$ to calculate our optimization objective called spatial diversity objective by maximizing the spatial entropy:

$$\mathcal{H}(\mathbf{p}(\mathbf{I})) = -\sum_{k=1}^{K} p_k(\mathbf{I}) \log p_k(\mathbf{I}) \quad (7)$$

**Wasserstein Term.** To encourage temporal smoothness between subsequent imaging decisions, we include a regularization term based on the entropic regularized Wasserstein distance (Sinkhorn distance) between imaging point sets at consecutive time steps. Given a previous set of imaging points $\mathbf{I}^{\text{prev}} = \{\mathbf{I}_1^{\text{prev}}, \ldots, \mathbf{I}_n^{\text{prev}}\}$ and current imaging points $\mathbf{I} = \{\mathbf{I}_1, \ldots, \mathbf{I}_n\}$, the cost matrix is defined as:

$$C_{ij} = \|\mathbf{I}_i - \mathbf{I}_j^{\text{prev}}\|^2. \tag{8}$$

Let $K = \exp(-C/\varepsilon)$ be the Gibbs kernel, where $\varepsilon > 0$ is the entropic regularization parameter. The optimal transport matrix $P^*$ is obtained via Sinkhorn iterations:

$$
\begin{aligned}
u^{(l+1)} &= \frac{\mu}{K v^{(l)}}, \\
v^{(l+1)} &= \frac{\nu}{K^\top u^{(l+1)}},
\end{aligned}
\tag{9}
$$

where $\mu = \nu = \frac{1}{n}\mathbf{1}_n$ are uniform marginals. The regularized Wasserstein distance is then:

$$\mathcal{W}_\varepsilon(\mathbf{I}, \mathbf{I}^{\text{prev}}) = \sum_{i,j} P_{ij}^* C_{ij}. \tag{10}$$

This term is scaled by a temporal step $\tau$ and regularization coefficient $\lambda$ in the full objective:

$$\mathcal{L}_{\text{JKO}} = -\mathcal{H}(\mathbf{p}(\mathbf{I})) + \frac{\lambda}{2\tau}\mathcal{W}_\varepsilon(\mathbf{I}, \mathbf{I}^{\text{prev}}). \tag{11}$$

**Optimization Problem.** The full scheduling problem is:

$$\min_{\mathbf{I}} \quad -\mathcal{H}(\mathbf{p}(\mathbf{I})) + \frac{\lambda}{2\tau}\mathcal{W}_\varepsilon(\mathbf{I}, \mathbf{I}^{\text{prev}}) \text{ s.t. } \mathbf{I}_i \in \mathcal{I}_i \tag{12}$$

## 2.2 Scanning Distribution Optimization

Our objective is to transform an initial distribution of imaging points into an optimized configuration that maximizes spatial coverage while maintaining temporal coherence. We formulate this as the formulated optimization problem (Eq. 12). We first give optimality of the entropy term (Proposition 3.1) and the Wasserstein term (Proposition 3.2). Second, we give the optimality of the optimization problem. Next, we propose a gradient-based algorithm to optimize the optimization problem. Last, we analyze the convergence behavior of optimization problem.

**Entropy Optimality.** The term $-\mathcal{H}(\mathbf{p}) = \sum_{k=1}^{K} p_k \log p_k$ penalizes low-diversity scanning patterns. A high-entropy distribution implies a uniform spread of imaging attention over the area $\mathcal{A}$, reducing redundancy and maximizing information gain. The optimality condition for this term leads to a repulsion effect among imaging points. We have the following proposition:

**Proposition 1.** Let a bounded region $\mathcal{A} \subset \mathbb{R}^2$ be partitioned into $K$ non-overlapping subregions $\{g_1, g_2, \ldots, g_K\}$ with respective areas $\{a_1, a_2, \ldots, a_K\}$ such that $\sum_{k=1}^{K} a_k = A$. Let $\mathbf{I} = \{I_1, \ldots, I_n\}$ be a imaging point set in $\mathcal{A}$, representing the relative observation density. Then the entropy

$$\mathcal{H}(\mathbf{I}) = -\sum_{j=1}^{K} q_j(\mathbf{I}) \log q_j(\mathbf{I}), \tag{13}$$

where

$$q_j(\mathbf{I}) = \frac{\sum_{i=1}^{n} \exp\left(-\frac{\|\mathbf{g}_j - \mathbf{I}_i\|^2}{2\sigma^2}\right)}{\sum_{k=1}^{K} \sum_{i=1}^{n} \exp\left(-\frac{\|\mathbf{g}_k - \mathbf{I}_i\|^2}{2\sigma^2}\right)}, \tag{14}$$

can be maximized. The proof of Proposition 1 is presented in Appendix A.1

**Wasserstein Optimality.** The second term in the loss, $\mathcal{W}_\varepsilon(\mathbf{I}, \mathbf{I}^{\text{prev}})$, corresponds to the entropy-regularized Wasserstein distance between the current and previous distributions. This term ensures temporal smoothness and discourages abrupt movements between successive frames. It acts as a

regularizer that maintains consistency in satellite behavior, crucial for respecting satellite slewing limits and minimizing mechanical stress. We have the following proposition:

**Proposition 2.** Let $\mathbf{I}^{\text{prev}} = \{I_1^{\text{prev}}, \dots, I_n^{\text{prev}}\}$ and $\mathbf{I}^{\text{new}} = \{I_1^{\text{new}}, \dots, I_n^{\text{new}}\}$ be two empirical distributions of $n$ imaging points in region $\mathcal{A} \subset \mathbb{R}^2$, each represented as uniform probability measures: $\mu = \frac{1}{n} \sum_{i=1}^{n} \delta_{I_i^{\text{prev}}}$ and $\nu = \frac{1}{n} \sum_{i=1}^{n} \delta_{I_i^{\text{new}}}$.

Let $W_\epsilon(\mu, \nu)$ denote the entropy-regularized Sinkhorn approximation of the squared 2-Wasserstein distance with regularization parameter $\epsilon > 0$. Then the Wasserstein cost term

$$\mathcal{W}(\mathbf{I}^{\text{prev}}, \mathbf{I}^{\text{new}}) = \sum_{i,j=1}^{n} \pi_{ij} \| I_i^{\text{prev}} - I_j^{\text{new}} \|^2, \tag{15}$$

where $\pi_{ij} \propto \exp\left(-\frac{\| I_i^{\text{prev}} - I_j^{\text{new}} \|^2}{\epsilon}\right)$, is minimized when

$$\mathbf{I}^{\text{new}} = \mathbf{I}^{\text{prev}}, \tag{16}$$

i.e., when the updated imaging points coincide with the initial configuration. In that case, the minimum Wasserstein distance is:

$$\mathcal{W}_{\min} = 0. \tag{17}$$

The proof of Proposition 2 is shown in Appendix A.2

**Theorem 1.** Let $\mathbf{I}^{\text{prev}} = \{I_1^{\text{prev}}, \dots, I_n^{\text{prev}}\}$ and let $\mathcal{H}(\mathbf{I})$ denote the soft entropy as defined in Equation equation 7. Define the following entropy-regularized variational problem:

$$\mathbf{I}^* = \arg \min_{\mathbf{I} \in \mathcal{A}^n} \left\{ \frac{1}{2\tau} \mathcal{W}(\mathbf{I}, \mathbf{I}^{\text{prev}}) - \mathcal{H}(\mathbf{I}) \right\}, \tag{18}$$

where $\mathcal{W}(\cdot, \cdot)$ denotes the entropy-regularized Wasserstein distance and $\tau > 0$ is the step size.

Then the solution $\mathbf{I}^*$ defines the most entropic distribution of imaging points within the feasible space $\mathcal{A}$, among all distributions that are close to the previous configuration $\mathbf{I}^{\text{prev}}$ in transport distance. We provide its proof in Appendix A.3.

**Convergence Behavior.** Empirically, the entropy term is concave in $\mathbf{p}$ but becomes non-convex when expressed in $\mathbf{I}$ due to the nonlinear mapping. However, thanks to the smoothness introduced by the Gaussian kernel and the entropy-regularized Sinkhorn approximation of Wasserstein distance, the overall loss is differentiable and sufficiently smooth. We propose Algorithm 1 which is the pseudo-code of scanning distribution optimization (SDO)). We can observe rapid convergence of the method, with the gradient norm consistently decreasing and entropy increasing. We have the following theorem to make sure the convergence:

---

**Algorithm 1:** Pseudo-code of SDO

**Input:** Initial imaging point distribution $\mathcal{I}$, grid set $\mathbf{G}$, number of iterations $T$
**Output:** Optimal distribution $\mathcal{I}^*$
**for** $t \leftarrow 1$ **to** $T$ **do**
    $\rho \leftarrow |\mathcal{I}|/|\mathbf{G}|$;
    **if** $\rho < \beta \frac{(\Delta_g)^2}{W_{imaging} \cdot H_{imaging}}$ **then**
        $\mathcal{I} \leftarrow \mathcal{I} \cup \mathbf{I}$;
    **else**
        **for** $m \leftarrow 1$ **to** $M_{iter}$ **do**
            Compute gradient $\nabla \mathcal{L}_{\text{JKO}}(\mathcal{I})$;
            Update
            $\mathcal{I} \leftarrow \mathcal{I} - \eta \cdot \nabla \mathcal{L}_{\text{JKO}}(\mathcal{I})$ s.t. (Eq. 3);
    $\mathcal{I}^* \leftarrow \mathcal{I}^* \cup \mathcal{I}$;
    $\mathcal{I} \leftarrow \phi$;

---

**Theorem 2.** Let $\{\mathbf{I}^{(k)}\}_{k=0}^{\infty}$ be the sequence of imaging point configurations generated by successive minimization of the JKO functional:

$$\mathbf{I}^{(k+1)} = \arg \min_{\mathbf{I} \in \mathcal{A}^n} \left\{ \frac{1}{2\tau} \mathcal{W}(\mathbf{I}, \mathbf{I}^{(k)}) - \mathcal{H}(\mathbf{I}) \right\}. \tag{19}$$

Then the iterative sequence $\mathbf{I}^{(k)}$ converges to a stationary point of the JKO objective, with a non-increasing energy functional and, in the limit $\tau \to 0$, recovers the continuous Wasserstein gradient flow of the entropy functional. The proof of Theorem 2 is shown in Appendix A.4

## 2.3 OPTIMAL MATCHING MODULE

After determining the optimal imaging point distribution $I_1, \ldots, I_n$, the system must assign each point to a suitable satellite such that all targets are imaged while minimizing each satellite's scanning effort, measured by angular deviation from its prior orientation. This task is formulated as a bipartite assignment problem between imaging points and satellite-time pairs. To quantify assignment cost, we define $c_{ij}$ as the total angular adjustment—i.e., the sum of pitch and roll changes—required for satellite $s_i$ to reach point $I_j$, relative to its prior orientation. To efficiently solve the matching problem, we adopt a hybrid strategy: (i) we first filter candidate assignments using a nearest-neighbor preselection based on angular proximity, and (ii) we then solve the reduced problem using the Hungarian algorithm to compute the globally optimal matching. This approach ensures that each imaging point is covered with minimal maneuvering overhead, maintaining high spatial efficiency while respecting mechanical constraints. The final output is a mapping $\mathcal{M} : I_j \mapsto (s_i, t_j)$, specifying which satellite executes which imaging point at what time. By integrating local filtering with global optimization, this module effectively bridges high-level distribution planning and low-level angle-aware scheduling.

## 2.4 CALCULATION OF CONTROLLING ANGLES

To execute the optimized imaging distribution $I_{i,t}$, we must compute the camera control angles (pitch $\theta^{\text{pitch}}i, t$ and roll $\theta^{\text{roll}}i, t$) that steer each satellite's field of view toward its assigned ground target. Given the satellite's position, altitude, and azimuth orientation, we solve an inverse problem: minimizing the geodetic error between the desired imaging point and the one predicted by the forward projection model. This yields a per-satellite control plan via efficient 2D optimization, ensuring that all targets are accurately covered within physical actuation limits. The angles serve as executable commands for onboard scanning systems, bridging high-level planning and deployment.

# 3 EXPERIMENTS

In this section, we comprehensively evaluate our proposed entropy-driven satellite scanning framework under realistic orbital conditions. Our experiments aim to answer three key questions: (1) How effectively does our method reduce observation delays and improve spatiotemporal coverage compared to existing baselines? (2) How does the scanning distribution adapt to different regions? (3) How well does the proposed optimal matching module translate high-level distributional plans into smooth and efficient satellite scanning trajectories?

## 3.1 EXPERIMENTAL SETTINGS

We evaluate our method using a trace-driven simulation framework based on real-world Starlink TLE data. Our experiments focus on three wildfire-prone U.S. states—California cal (2025), Colorado col (2025), and Texas tex (2025)—and assess both static monitoring performance and dynamic responsiveness under realistic orbital visibility constraints. For each region, we simulate satellite visibility and task scheduling starting at `2025-01-10 00:00:00 UTC`, over a 200-minute observation window. The area is discretized into a uniform spatial grid with a resolution of approximately $10 \times 10\,\text{km}^2$ per cell. For each grid cell, we record the timestamps of all feasible imaging events that satisfy field-of-view and slewing constraints. The **time gap** is defined as the longest interval between any two consecutive observations of a cell. We use the maximum time gap across all cells as our primary delay metric, as wildfire surveillance is highly sensitive to worst-case blind spots, where prolonged observation gaps may allow fires to spread undetected.

**Metrics.** To quantitatively evaluate our method, we adopt two metrics that capture complementary aspects of spatiotemporal coverage quality. The definitions of the two metrics are presented as follows. **(i) Maximum Revisit Gap.** To evaluate long-term temporal consistency, we track the time intervals between successive observations of each cell over a fixed horizon. For each cell, we compute the maximum time gap between consecutive scans. The overall metric is the maximum of these per-cell values. This captures the worst-case staleness and reflects how well the system avoids prolonged blind spots. **(ii) Delay Distribution.** This metric captures the spatial distribution of observation continuity across the region by measuring the *Maximum Revisit Gap* for each grid cell. We visualize the distribution as a heatmap, where ideally, all cells exhibit similar and short revisit intervals. This metric evaluates the system's ability to avoid localized blind spots and provide consistent temporal coverage across the entire region.

Together, these metrics provide a comprehensive assessment of both the responsiveness and persistence of our observation scheduling framework.

**Baselines.** We compare our method against three representative baselines that reflect standard strategies for satellite-based scanning: **(i) Push Broom Scanner Keto & Watters (2025).** A traditional nadir-pointing policy where satellites follow their fixed polar orbits and capture images directly beneath them at regular intervals. This strategy reflects the behavior of many commercial platforms such as Planet Labs pla (2025) and does not optimize for coverage latency. **(ii) Whisk Broom Scanner Du et al. (2025).** A configuration-aware variant where satellites follow a Walker constellation design, aiming for uniform longitudinal coverage. Like the push broom scanner, this approach does not adapt to surface dynamics or prioritize unvisited regions. **(iii) 4-Wide Swath Langer et al. (2024).** A capture mode that extends a satellite's across-track coverage by alternately pointing and scanning in the positive and negative across-track directions with respect to a target location. These baselines allow us to evaluate the effectiveness of our proposed framework under realistic and competitive scheduling conditions.

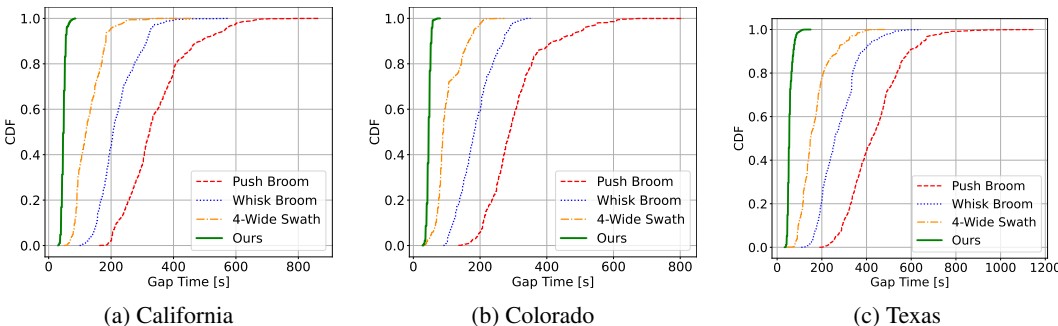

|           (a) California           |           (b) Colorado           |           (c) Texas           |

Figure 4: Cumulative distribution functions (CDFs) of time gap across three U.S. states, comparing our proposed method against three mainstream baselines. (a) California: our median gap is 81 s, a reduction of 519 s (86.5 %) and 776 s (90.6 %) relative to the existing methods at 600 s and 857 s, respectively. (b) Colorado: our median gap is 78 s, outperforming the 380 s and 800 s existing methods by 302 s (79.5 %) and 722 s (90.3 %). (c) Texas: our median gap is 178 s, improving upon the 620 s and 1 178 s existing methods by 442 s (71.3 %) and 1 000 s (84.8 %).

## 3.2 COMPARISON WITH BASELINES

To quantify the quality of spatial coverage in the absence of wildfire events, we run our method and three baselines in static monitoring mode over a 200-minute time window. The primary metric is the **inter-observation gap**, the time interval between consecutive observations for each grid cell.

**CDF of Time Gap** We compute the cumulative distribution function (CDF) of cell-wise observation gaps, where a left-shifted curve indicates more frequent observations and reduced monitoring delays. Figure 4 compares our entropy-based scheduling framework with three traditional baselines across California, Colorado, and Texas. Our method achieves substantial reductions in observation gaps—up to 90.6%, 90.3%, and 84.8% in the respective regions—highlighting its ability to significantly enhance revisit frequency. Furthermore, the smoother shape of our CDF curves demonstrates improved spatial uniformity in observation coverage.

**Delay Distribution Map.** We visualize the spatial distribution of maximum observation delays using a heatmap, where each grid cell is colored according to the longest time gap between any two successive observations at that location. This provides a clear indication of both well-covered and under-observed areas. As illustrated in Figure 5, our method substantially reduces the maximum revisit delay across all evaluated regions, consistently outperforming the three baseline strategies.

## 3.3 SATELLITE-IMAGING MATCHING RESULTS

To evaluate the effectiveness of our *Optimal Matching Module*, we analyze how well the optimized imaging points are assigned to available satellites to produce a coherent and efficient scanning plan. After computing the most informative imaging point distribution via the entropy-driven optimization, the matching module allocates each point to a specific satellite-time pair while minimizing the total angular control effort. Figure 6 visualizes the resulting scanning trajectories of the satellites over three regions (California, Colorado and Texas). Each trajectory illustrates how a satellite

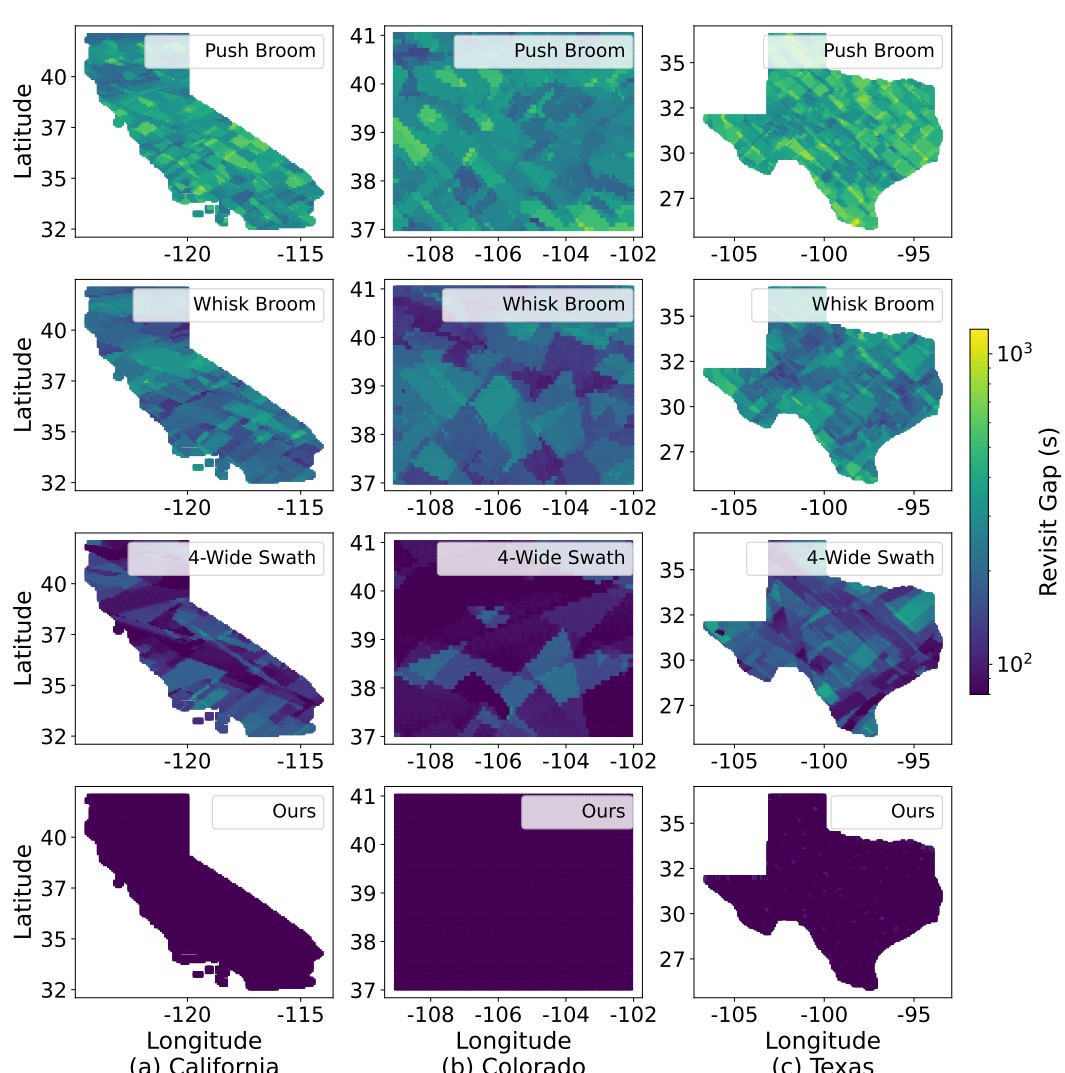

Figure 5: Map-based spatial distribution of time gaps for the different test maps. Colors represent the revisit gap in seconds, and the color bar is plotted on a logarithmic scale to emphasize variations across orders of magnitude. (a) California. (b) Colorado. (c) Texas.

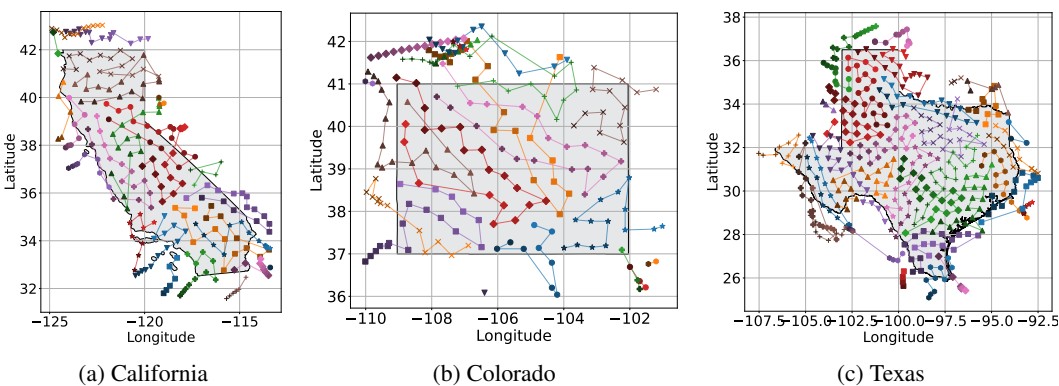

Figure 6: Matching maps after optimal distribution of imaging points to satellites for three regions: (a) California, (b) Colorado, (c) Texas. Each point is color-coded by the satellite it has been assigned to under the optimization, illustrating the scanning strategies across the satellite constellation.

moves across its assigned imaging targets, reflecting the outcome of the hybrid matching algorithm that balances spatial assignment and maneuver smoothness.

## 4 RELATED WORK

**Earth Observation.** Recent advances in computer vision have enabled high-accuracy Earth observation Pan et al. (2022); Hu et al. (2023); Tuncel et al. (2023) and reduced data transmission between satellites and ground stations Zhang et al. (2024b). These models leverage deep architectures to identify visual patterns of smoke, heat, or structural changes, offering new capabilities for rapid hazard detection. However, they typically operate under the assumption that imagery streams are frequent, uniformly distributed, and latency-tolerant — conditions that real-world satellite systems rarely satisfy, especially over large or dynamic regions. To address observation limitations, multimodal systems Fu et al. (2024); Dritsas & Trigka (2025); Abdellatif et al. (2025) have incorporated static ground sensors and UAVs to complement satellite views, achieving finer geo-location and real-time alerting. Yet such infrastructures are inherently constrained by deployment cost, maintenance, and limited spatial coverage, making them impractical for continental-scale or global monitoring.

Our work complements these downstream analysis efforts by tackling the upstream acquisition problem: ensuring that satellite sensors are directed toward high-utility regions in a timely and balanced manner. Unlike traditional systems that treat data acquisition as an uncontrollable background process, we treat observation scheduling itself as a first-class optimization target, designing a scalable, dynamic framework for proactive, near real-time Earth observation.

**LEO Satellite Systems.** The proliferation of Low Earth Orbit (LEO) constellations has driven research in networking, topology design, routing optimization, and emulation frameworks Li et al. (2023); Zhang et al. (2024a); Infantes et al.; Li et al. (2021); Bhattacherjee et al. (2018); Qin et al. (2024); Lai et al. (2023); Xing et al. (2024), largely treating satellites as passive relays to optimize data transmission. Application studies further integrate LEO with IoT and web services Ren et al. (2024); Shenoy et al. (2024); Li et al. (2024); Shayea et al. (2024), yet satellites remain transparent links rather than reasoning agents. Observation-focused work has modeled energy-constrained single-satellite schedules Wen et al. (2023); Mercado-Martínez et al. (2025) or prioritized image downlink Tao et al. (2024), but assumes imagery is already captured. In contrast, we reframe satellites as *active sensing agents* that proactively decide when and where to observe, enabling coordinated multi-agent sensing to maximize spatial coverage and minimize observation gaps.

**Active Vision.** Active vision research conceptualizes perception as an adaptive, goal-driven process in which sensing agents actively choose viewpoints to maximize information gain Aloimonos et al. (1988); Ballard (1991). Early work introduced purposive and behavioral strategies Aloimonos (1992); Ballard & Brown (1992), later extended by reinforcement learning and information-theoretic approaches, including uncertainty-aware view selection policies Gallos & Ferrie (2019); Martin (2006); Xu & Luger (2007). Building on this line of research, our work departs in three key ways: we optimize a continuous spatial scanning distribution directly from information-theoretic objectives rather than learning reactive policies; we scale beyond local camera or robotic navigation to large-area orbital systems with globally coupled sensing actions; and we propose a novel entropy-based framework with Wasserstein regularization that jointly maximizes information gain and spatial fairness. In doing so, we extend active vision principles toward a scalable and principled formulation for satellite-based Earth observation.

## 5 CONCLUSION

In this paper, we presented an entropy-driven framework for near real-time Earth observation with large-scale LEO constellations. Our formulation casts scanning optimization as a Wasserstein gradient flow over imaging point distributions, where spatial entropy maximization promotes diverse and fair coverage while Wasserstein regularization ensures smooth evolution from initial scan plans. To operationalize this principle, we introduced a differentiable solver that maps optimized imaging points into physically executable camera angles, together with an efficient satellite-to-task assignment module that minimizes slewing effort via a hybrid of the Hungarian algorithm and nearest-neighbor filtering. Through large-scale evaluations on real-world Starlink trajectories, our proposed framework achieves full-region coverage within minutes and provides up to 10× faster scanning compared to conventional orbit-based strategies, significantly reducing observation latency and improving regional monitoring efficiency.

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

## A MATHEMATICAL DERIVATIONS

### A.1 PROOF OF PROPOSITION 1

We aim to show that the entropy

$$\mathcal{H}(\mathbf{I}) = -\sum_{j=1}^{K} q_j(\mathbf{I}) \log q_j(\mathbf{I}), \tag{20}$$

is maximized when the soft assignment probability $q_j(\mathbf{I})$ is proportional to the area $a_j$ of grid cell $g_j$, that is, $q_j = \frac{a_j}{A}$ where $A = \sum_{j=1}^{K} a_j$.

Recall the definition:

$$q_j(\mathbf{I}) = \frac{\sum_{i=1}^{n} \exp\left(-\frac{\|\mathbf{g}_j - \mathbf{I}_i\|^2}{2\sigma^2}\right)}{\sum_{k=1}^{K} \sum_{i=1}^{n} \exp\left(-\frac{\|\mathbf{g}_k - \mathbf{I}_i\|^2}{2\sigma^2}\right)}. \tag{21}$$

Let us denote:

$$s_j = \sum_{i=1}^{n} \exp\left(-\frac{\|\mathbf{g}_j - \mathbf{I}_i\|^2}{2\sigma^2}\right), \tag{22}$$

$$Z = \sum_{k=1}^{K} s_k = \sum_{k=1}^{K} \sum_{i=1}^{n} \exp\left(-\frac{\|\mathbf{g}_k - \mathbf{I}_i\|^2}{2\sigma^2}\right), \tag{23}$$

so that $q_j = s_j/Z$. The entropy becomes:

$$\mathcal{H}(\mathbf{I}) = -\sum_{j=1}^{K} \frac{s_j}{Z} \log \frac{s_j}{Z} = -\sum_{j=1}^{K} \frac{s_j}{Z} \left(\log s_j - \log Z\right) \tag{24}$$

$$= -\sum_{j=1}^{K} \frac{s_j}{Z} \log s_j + \log Z \sum_{j=1}^{K} \frac{s_j}{Z} = -\sum_{j=1}^{K} \frac{s_j}{Z} \log s_j + \log Z. \tag{25}$$

Therefore, maximizing $\mathcal{H}(\mathbf{I})$ is equivalent to maximizing the expression:

$$\log Z - \sum_{j=1}^{K} \frac{s_j}{Z} \log s_j. \tag{26}$$

Now assume that $s_j$ is proportional to the area of grid cell $g_j$, i.e.,

$$s_j = n \cdot \frac{a_j}{A} \quad \text{for all } j = 1, \ldots, K. \tag{27}$$

Then the normalization constant becomes:

$$Z = \sum_{j=1}^{K} s_j = \sum_{j=1}^{K} n \cdot \frac{a_j}{A} = n. \tag{28}$$

Thus,

$$q_j = \frac{s_j}{Z} = \frac{n \cdot \frac{a_j}{A}}{n} = \frac{a_j}{A}, \tag{29}$$

which matches the area-normalized target distribution. Plugging back into the entropy expression yields:

$$\mathcal{H}_{\max} = -\sum_{j=1}^{K} \frac{a_j}{A} \log \frac{a_j}{A}. \tag{30}$$

Hence, the entropy $\mathcal{H}(\mathbf{I})$ is maximized when the imaging point distribution induces a grid-level coverage proportional to the area of each cell.

## A.2 PROOF OF PROPOSITION 2

We consider the entropy-regularized optimal transport problem between $\mu$ and $\nu$ with cost matrix $C_{ij} = \|I_i^{\text{prev}} - I_j^{\text{new}}\|^2$:

$$\min_{\pi \in \Pi(\mu,\nu)} \sum_{i,j} \pi_{ij} C_{ij} + \epsilon \sum_{i,j} \pi_{ij}(\log \pi_{ij} - 1), \tag{31}$$

where $\Pi(\mu, \nu)$ denotes the set of joint couplings with uniform marginals:

$$\sum_j \pi_{ij} = \frac{1}{n}, \quad \sum_i \pi_{ij} = \frac{1}{n}. \tag{32}$$

Suppose $\mathbf{I}^{\text{new}} = \mathbf{I}^{\text{prev}}$. Then for all $i$,

$$C_{ii} = 0, \quad C_{ij} > 0 \quad \text{for } i \neq j.$$

This makes the exponential kernel $K_{ij} = \exp(-C_{ij}/\epsilon)$ maximized on the diagonal, i.e.,

$$K_{ij} = \begin{cases} 1, & i = j, \\ < 1, & i \neq j. \end{cases}$$

Therefore, the optimal plan $\pi^*$ assigns all mass on the diagonal:

$$\pi_{ij} = \begin{cases} \frac{1}{n}, & i = j, \\ 0, & i \neq j. \end{cases}$$

Substituting into the cost:

$$\mathcal{W}_\epsilon(\mu, \mu) = \sum_{i=1}^n \frac{1}{n} \cdot 0 = 0.$$

Any deviation $\mathbf{I}^{\text{new}} \neq \mathbf{I}^{\text{prev}}$ will increase some $C_{ij} > 0$, leading to a strictly positive cost. Hence, the minimum is uniquely achieved at identity matching, i.e., when $\mathbf{I}^{\text{new}} = \mathbf{I}^{\text{prev}}$.

## A.3 PROOF OF THEOREM 1

We aim to minimize the following objective:

$$\mathcal{F}(\mathbf{I}) := \frac{1}{2\tau} \mathcal{W}(\mathbf{I}, \mathbf{I}^{\text{prev}}) - \mathcal{H}(\mathbf{I}). \tag{33}$$

**Step 1: Convexity of the objective.**
The entropy functional $\mathcal{H}(\mathbf{I})$ is strictly concave in $\mathbf{I}$ due to the log-sum-exp structure of softmax probabilities. Its negative $-\mathcal{H}(\mathbf{I})$ is therefore convex.

The entropy-regularized Wasserstein distance $\mathcal{W}(\mathbf{I}, \mathbf{I}^{\text{prev}})$ is also convex in $\mathbf{I}$ for fixed $\mathbf{I}^{\text{prev}}$, due to the convexity of the transport cost and the log-sum-exp form of the Sinkhorn regularization Cuturi (2013).

Hence, the full functional $\mathcal{F}(\mathbf{I})$ is convex.

**Step 2: Existence of a minimizer.**
Since $\mathcal{A}$ is a compact subset of $\mathbb{R}^2$ and $\mathbf{I} \in \mathcal{A}^n$, the feasible set is compact. The objective $\mathcal{F}(\mathbf{I})$ is lower semicontinuous and coercive due to the Wasserstein term. Therefore, by the Weierstrass theorem, a minimizer $\mathbf{I}^*$ exists.

**Step 3: Optimality condition.**
At optimality, we have:

$$\mathbf{I}^* = \arg\min_{\mathbf{I} \in \mathcal{A}^n} \left\{ \frac{1}{2\tau} \mathcal{W}(\mathbf{I}, \mathbf{I}^{\text{prev}}) - \mathcal{H}(\mathbf{I}) \right\}, \tag{34}$$

which implies that among all feasible distributions $\mathbf{I}$ close to $\mathbf{I}^{\text{prev}}$ (in terms of $\mathcal{W}$), the one with highest entropy is selected. This provides a balance between exploration (entropy) and consistency with prior state (transport).

**Step 4: Gradient flow interpretation.**
(Eq. 18) corresponds to one step of the Jordan–Kinderlehrer–Otto (JKO) scheme for Wasserstein gradient flows Jordan et al. (1999), discretized in time:

$$\mathbf{I}^{(k+1)} = \arg\min_{\mathbf{I}} \left\{ \frac{1}{2\tau} \mathcal{W}(\mathbf{I}, \mathbf{I}^{(k)}) - \mathcal{H}(\mathbf{I}) \right\}. \tag{35}$$

Hence, $\mathbf{I}^*$ is the result of one variational update along the Wasserstein gradient flow of the entropy functional, projected to the feasible region $\mathcal{A}^n$.

### A.4 Proof of Theorem 2

We proceed in three parts:

**(1) Monotonic Descent of Energy Functional.** By definition of each JKO update, we have:

$$\mathbf{I}^{(k+1)} = \arg\min_{\mathbf{I}} \left\{ \frac{1}{2\tau} \mathcal{W}(\mathbf{I}, \mathbf{I}^{(k)}) - \mathcal{H}(\mathbf{I}) \right\}. \tag{36}$$

Thus, evaluating at $\mathbf{I} = \mathbf{I}^{(k)}$ gives:

$$\mathcal{F}(\mathbf{I}^{(k+1)}) \leq \mathcal{F}(\mathbf{I}^{(k)}). \tag{37}$$

This implies that the energy functional is non-increasing across iterations.

**(2) Existence of Convergent Subsequence.** Because the feasible domain $\mathcal{A}^n$ is compact and the functional is coercive (the Wasserstein term acts as a regularizer), we have that the sequence $\{\mathbf{I}^{(k)}\}$ admits at least one accumulation point. Furthermore, because the objective is convex and lower semi-continuous (as shown in Theorem 2.2), the sequence converges to a stationary point $\mathbf{I}^*$ that satisfies the optimality condition:

$$0 \in \partial \left( \frac{1}{2\tau} \mathcal{W}(\cdot, \mathbf{I}^*) - \mathcal{H}(\cdot) \right).$$

**(3) Wasserstein Gradient Flow Limit.** When $\tau \to 0$, the JKO scheme becomes a time-discretized approximation of a continuous gradient flow. Specifically, from the theory of Wasserstein gradient flows Jordan et al. (1999); Ambrosio et al. (2008), we know:

$$\lim_{\tau \to 0} \frac{\mathbf{I}^{(k+1)} - \mathbf{I}^{(k)}}{\tau} = \nabla_{\mathcal{W}} \mathcal{H}(\mathbf{I}^{(k)}), \tag{38}$$

which is the defining evolution equation for the Wasserstein gradient flow:

$$\partial_t \mathbf{I}(t) = \nabla_{\mathcal{W}} \mathcal{H}(\mathbf{I}(t)).$$

Hence, the sequence $\{\mathbf{I}^{(k)}\}$ approximates a discretized solution to this gradient flow in $\mathcal{P}_2(\mathcal{A})$, the space of probability measures with finite second moments.

## B Implementation Details

### B.1 Hardware Figuration

All experiments are conducted on a single NVIDIA RTX 2060 GPU with 6GB memory and an AMD RYZEN 7 4000 CPU with 8 cores and 16 threads.

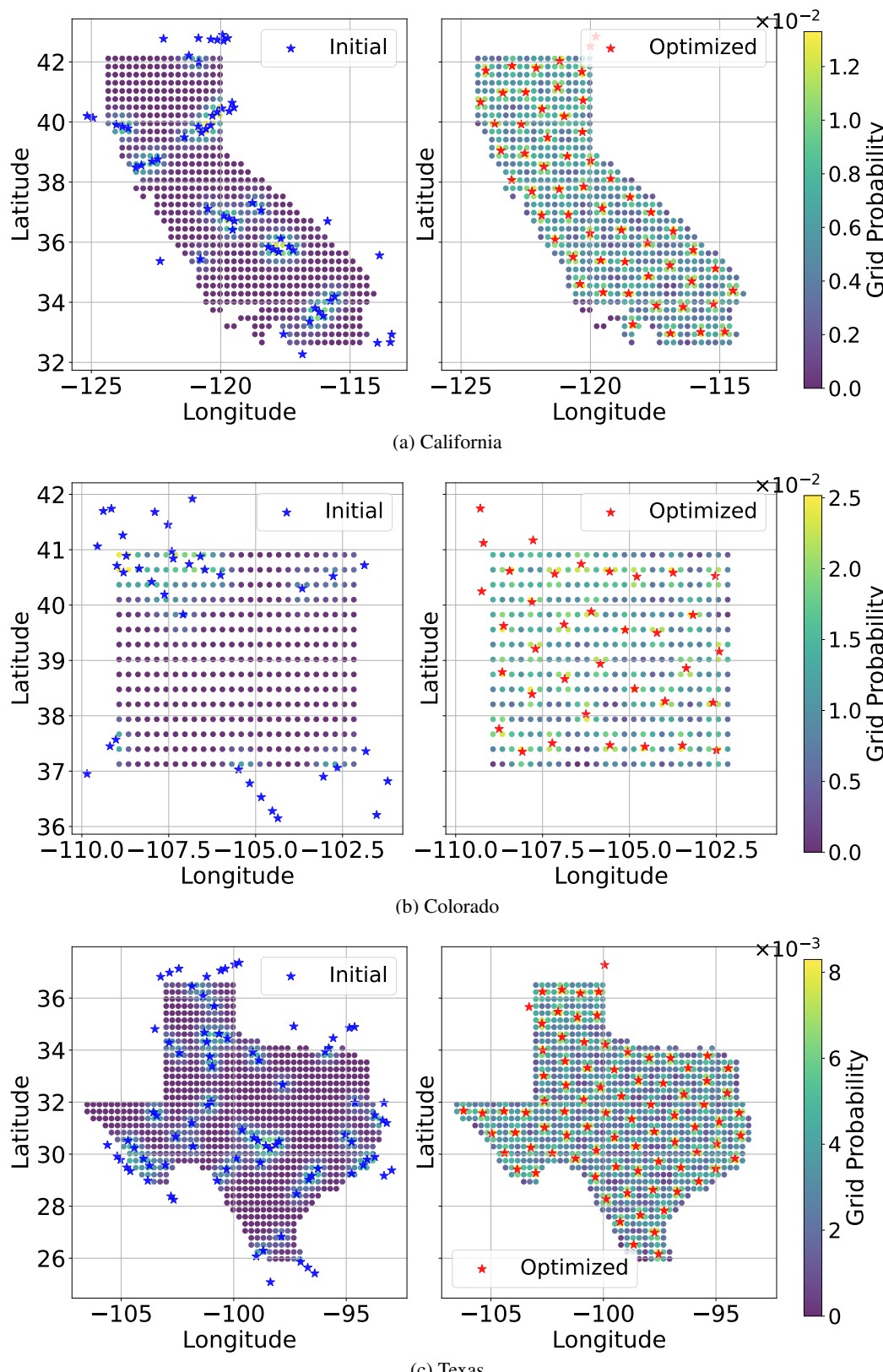

(a) California

(b) Colorado

(c) Texas

Figure 7: Grid probability distributions. Blue pentagrams indicate the initial imaging point distribution; red pentagrams denote the optimized imaging points obtained from our scheduling framework.

Table 1: Parameter Settings

| Region | Method | Grid Size (km $\times$ km) | $\beta$ | $M_{iter}$ | $\eta$ |
|---|---|---|---|---|---|
| California | Push Broom | $25 \times 25$ | - | - | - |
| | Whisk Broom | $25 \times 25$ | - | - | - |
| | 4 Wide Swath | $25 \times 25$ | - | - | - |
| | Ours | $25 \times 25$ | 1.1 | 100 | 0.0015 |
| Colorado | Push Broom | $20 \times 20$ | - | - | - |
| | Whisk Broom | $20 \times 20$ | - | - | - |
| | 4 Wide Swath | $20 \times 20$ | - | - | - |
| | Ours | $20 \times 20$ | 1.0 | 120 | 0.0018 |
| Texas | Push Broom | $30 \times 30$ | - | - | - |
| | Whisk Broom | $30 \times 30$ | - | - | - |
| | 4 Wide Swath | $30 \times 30$ | - | - | - |
| | Ours | $30 \times 30$ | 1.3 | 150 | 0.001 |

## B.2 PARAMETER SETTINGS

We evaluate our method across three geographically diverse U.S. states: California, Colorado, and Texas. For each region, we configure the grid resolution and algorithmic hyperparameters to reflect differences in spatial extent and satellite visibility patterns. Table 1 summarizes the specific parameter settings.

We use a uniform grid size per region, selected to balance granularity with computational cost. California and Colorado adopt $25 \times 25$ and $20 \times 20$ km$^2$ grids respectively to match their sizes and orbital density, while Texas, due to its larger area, uses a coarser $30 \times 30$ km$^2$ grid.

Baseline methods—Push Broom, Whisk Broom, and 4 Wide Swath—are deterministic scanning strategies and do not involve any tunable optimization hyperparameters. Therefore, parameters like learning rate $\eta$, entropy balance factor $\beta$, and maximum iteration count $M_{iter}$ are marked as not applicable.

In contrast, our method introduces three region-specific optimization parameters: (i)$\beta$ controls the balance between entropy maximization and revisit fairness; (ii) $M_{iter}$ denotes the number of optimization iterations used for convergence; (iii) $\eta$ is the learning rate for our gradient-based update mechanism.

We tune these values per region to account for differences in orbital density, area shape, and task complexity, ensuring robust convergence and high-quality scheduling.

## C ADDITIONAL EMPIRICAL RESULTS

### C.1 GRID MAP ANALYSIS

To better understand how our optimization framework evolves the observation distribution over time, we conduct a detailed analysis of the intermediate scheduling states using a grid map representation. Specifically, we discretize the target region into uniform spatial cells and visualize the probability mass assigned to each cell throughout the optimization process. As shown in Figure 7, this analysis serves two purposes. First, it allows us to verify that the optimization indeed promotes spatially diverse and balanced coverage as intended by the hybrid objective. Second, it provides empirical insights into how our method adaptively steers the distribution towards high-utility regions while maintaining fairness constraints.

Figure 7 illustrates the spatial distribution of soft grid probabilities before and after optimization across three representative regions—California, Colorado, and Texas. In the pre-optimization state, the distribution is highly concentrated: a few grid cells dominate the probability mass while most others receive near-zero probability. This indicates a localized, unbalanced initial configuration. After applying our Wasserstein entropy optimization, the probability mass becomes much more evenly distributed, with all grid cells receiving comparable values. This transition confirms that the optimization procedure spreads observation effort more uniformly across the region while respecting geometric and operational constraints. By examining the temporal evolution of the grid map, we

qualitatively validate the effectiveness and interpretability of our Wasserstein flow-based scheduling dynamics.

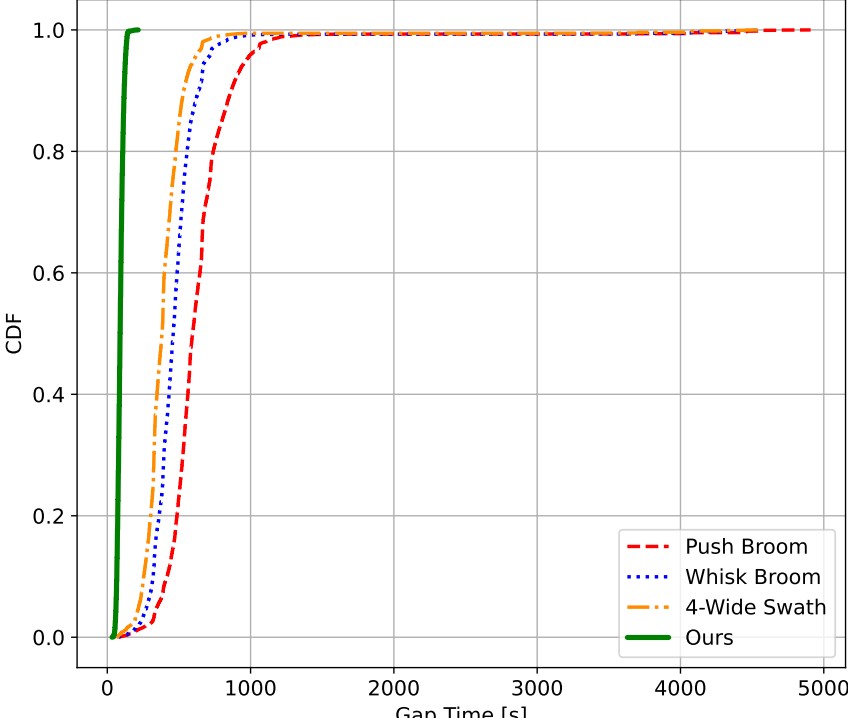

Figure 8: Cumulative distribution functions (CDFs) of South America.

### C.2    Scaling to Continent-Level Observation: South America

To evaluate the scalability and stability of our entropy-driven optimization under large-area constraints, we apply our method to a continent-scale setting: **South America**. This setting tests whether our scheduling strategy remains effective when the target region spans vast and heterogeneous terrain, introducing increased orbital diversity and visibility variability.

We discretize the landmass of South America into $30 \times 30$ km$^2$ grid cells and generate candidate imaging tasks based on real-world Starlink orbital predictions. Our method jointly optimizes the soft assignment entropy objective and applies Wasserstein-regularized JKO updates to maintain spatial diversity.

Despite the geographic scale, our approach achieves full-region coverage in under 12 minutes and maintains bounded maximum revisit gaps across all cells. The spatial distribution of imaging points remains balanced, with no regions disproportionately neglected. We observe that the Wasserstein gradient flow stabilizes the evolution of the scheduling distribution, avoiding collapse into high-density clusters and promoting smooth geographic spread.

As shown in Figure 8 and Figure 9, these results demonstrate that our method generalizes beyond regional and country-level applications, retaining its core advantages—spatial fairness, adaptive scheduling, and low-latency coverage—even when scaling to an entire continent. It further highlights the viability of real-time observation planning at planetary scale when paired with dense LEO constellations.

## D    Broader Impacts

This work contributes a novel scanning framework for near real-time Earth observation using large-scale LEO satellite constellations. By improving the efficiency and adaptability of satellite-based sensing, our method has the potential to support a wide range of societally beneficial applications,

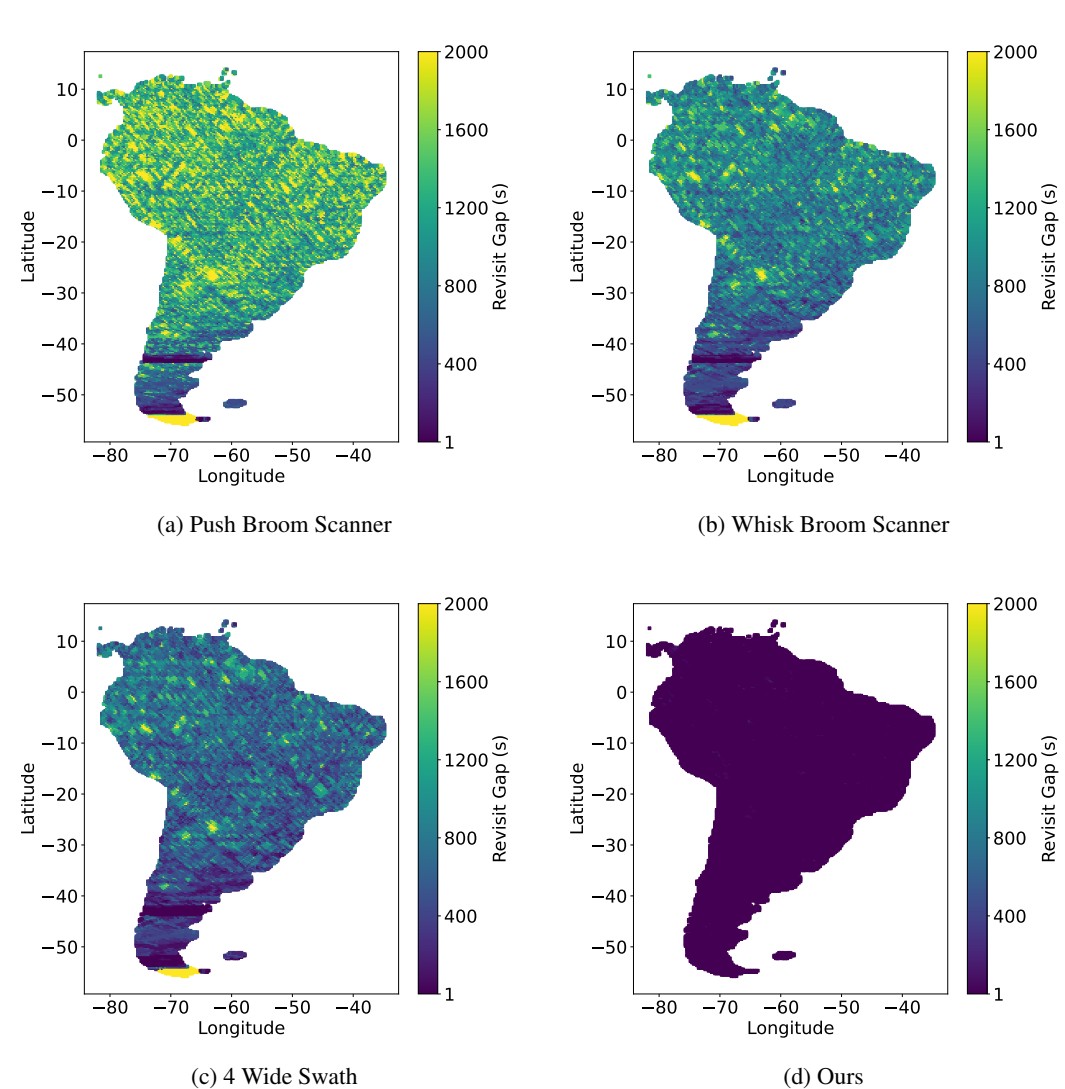

(a) Push Broom Scanner

(b) Whisk Broom Scanner

(c) 4 Wide Swath

(d) Ours

Figure 9: Map-based spatial distribution of time gaps for South America.

including early wildfire detection, environmental monitoring, disaster response, and agricultural planning. In particular, reducing the delay and spatial imbalance in satellite coverage can enable faster and more equitable access to critical Earth data, especially in regions vulnerable to climate-driven hazards.

From a technical perspective, our framework offers a generalizable approach to structured decision-making over constrained and dynamic spatial systems. Beyond satellite sensing, similar principles may apply to drone fleets, mobile sensor networks, or planetary-scale infrastructure monitoring.

However, as with any remote sensing system, the deployment of large-scale observation capabilities raises ethical considerations related to privacy, surveillance, and dual-use. While our work focuses on coarse-grained, environmental-scale monitoring (e.g., regional fire detection), we caution that such technologies could be repurposed for high-resolution or persistent surveillance. We recommend that real-world deployments follow transparent data policies and involve oversight from multidisciplinary stakeholders, including environmental scientists, civil agencies, and privacy advocates.

We are not aware of any direct negative societal consequences stemming from this research, and we have made no use of sensitive, personal, or private data during development. The proposed methods are intended to serve humanitarian and scientific goals.

