# OpenReview forum: "Entropy-Driven Scanning Optimization for Near Real-Time Earth Observation"
_ICLR.cc/2026/Conference — Submitted to ICLR 2026_

### Official Review · Reviewer_98HF · 2025-10-29

**Soundness:** 3
**Presentation:** 1
**Contribution:** 3
**Rating:** 6
**Confidence:** 2

**Summary:**

The paper introduces an Entropy-Driven Scanning Optimization (EDSO) framework designed to coordinate large Low Earth Orbit (LEO) satellite constellations for achieving near real-time Earth observation. The central idea is formulated as a continuous optimization problem that maximizes the spatial entropy of the imaging point distribution, thereby ensuring diverse, balanced, and non-redundant coverage across the target region. To preserve temporal coherence and reduce the satellites’ physical slewing effort, the objective function is regularized using the entropic Wasserstein distance (Sinkhorn distance) between consecutive scan plans. The complete pipeline incorporates a differentiable solver that transforms optimized ground points into executable camera control angles and a hybrid task-assignment algorithm combining the Hungarian method with a nearest-neighbor strategy. Experimental results demonstrate that the proposed approach achieves up to a tenfold speedup over conventional orbit-based strategies, enabling full-region scanning within minutes.

**Strengths:**

1. It presents the known entropy-based, information-theoretic formulation for large-scale LEO satellite scanning optimization. Framing the observation scheduling task as a continuous optimization problem over a spatial probability distribution constitutes a novel perspective, enabling the seamless integration of advanced mathematical frameworks such as Optimal Transport theory.
2. The manuscript features a comprehensive, full-stack pipeline, demonstrating not just a theoretical objective but also a practical implementation that includes differentiable geometry and efficient task assignment.

**Weaknesses:**

1. On page 2, line 86, the authors state "In Section 2, we survey relevant research''. However, Section 2 immediately proceeds to the methodological exposition, omitting any substantive review of prior literature. Moreover, several symbols within the equations are insufficiently defined, e.g., the $\sigma$ in Eq. (5) and the terms $\mathbf{g}_j $, $\mathbf{I}_i$ in Eq. (6).
2. The experimental evaluation is a major weak point. The comparison is made against traditional, non-optimized scanning strategies ("Push Broom," "Whisk Broom," "4-Wide Swath"). A rigorous comparison requires including modern, state-of-the-art scheduling and optimal control baselines.
3. The description of the hybrid matching strategy (nearest-neighbor preselection followed by the Hungarian algorithm) lacks sufficient implementation details, making it difficult to evaluate its efficiency and performance–cost trade-off.

**Questions:**

1. What is the measured wall-clock time for one time step's computation on the hardware used?
2. What quantitative particulars delineate the nearest-neighbor preselection within the Optimal Matching Module?

---

> ### Author Response · Authors · 2025-12-03
>
> ## Dear Reviewer 98HF,
>
> First, we sincerely thank you for your careful review and insightful feedback. Your comments have been invaluable in improving the quality of our work.
>
> ## Response to Weakness 1:
>
> **1. Correction of Structural Description**: We have corrected the sentence to accurately reflect the structure: "In Section 2, we present our proposed methodology. In Section 3, we present experimental
> verification and result comparisons. Relevant research is surveyed in Section 4. In Section 5, we conclude this paper"
>
> **2. Clarification of Mathematical Symbols**: We have added explicit definitions for the symbols in Eq. (5) and Eq. (6) to improve clarity. (i) $\sigma$ (Eq. 5): Represents the bandwidth (standard deviation) of the Gaussian kernel, which controls the "softness" or influence radius of each imaging point on the probability grid. (ii) $g_j$ (Eq. 6): $g_j \in \mathbb{R}^2$ is the spatial coordinate (center) of the $j$-th grid cell. (iii) $I_i \in \mathbb{R}^2$ is the spatial coordinate of the $i$-th imaging point. (iv) $d_{ji} = ||g_j - I_i||^2$ represents the squared Euclidean distance between the grid cell center and the imaging point.
>
> ## Response to Weakness 2:
> We agree that a rigorous comparison requires evaluating our method against modern, adaptive scheduling baselines. To address this, we have expanded our experiments to include two distinct strategies: (i) Genetic Algorithm (GA): A representative meta-heuristic method, serving as a strong modern optimization baseline. (ii) Greedy Strategy (Max-Gap): A heuristic baseline that dynamically assigns satellites to the grid cell with the longest time since the last observation.
>
> | Method | **California** <br> Median Gap (s) $\downarrow$ | **Colorado** <br> Median Gap (s) $\downarrow$ | **Texas** <br> Median Gap (s) $\downarrow$ |
> | :--- | :---: | :---: | :---: |
> | Push Broom | 857 | 800 | 1178 |
> | Whisk Broom | 600 | 380 | 620 |
> | 4-Wide Swath | 436 | 300 | 510 |
> | **Greedy (Max-Gap)** | **420** | **315** | **480** |
> | **Genetic Algorithm** | **243** | **216** | **303** |
> | **Ours (Entropy-Driven)** | **81** | **78** | **178** |
>
> ## Response to Weakness 3:
> We have revised Section 2.3 to accurately describe our "Hybrid Matching Strategy" as a two-stage decomposition that combines global optimal assignment with local heuristic sequencing. This approach effectively bridges the gap between high-level distribution planning and low-level execution. The strategy operates in two distinct phases:
>
> **Phase 1**: Global Assignment (Hungarian Algorithm) We first construct a cost matrix $C$ where $C_{ij}$ represents the Haversine distance between satellite $i$ and imaging target $j$. We then apply the Hungarian Algorithm (Linear Sum Assignment) to solve the global bipartite matching problem:
> $$\min \sum_{i,j} C_{ij} x_{ij} \quad \text{s.t.} \quad \sum_j x_{ij} = 1, \sum_i x_{ij} = 1$$
> This step ensures the globally optimal allocation of resources, minimizing the aggregate distance between satellites and targets.
>
> **Phase 2**: Sequence Refinement (Heuristic TSP) Once a set of targets $\mathcal{T}_i$ is assigned to satellite $i$, we determine the optimal execution order to minimize slewing effort. Since finding the exact optimal path is an NP-hard Traveling Salesman Problem (TSP), we employ a fast hybrid heuristic: (i) Nearest Neighbor (NN): Generates an initial greedy path by iteratively selecting the closest unvisited target. (ii) 2-Opt Local Search: Iteratively improves the path by reversing segments to eliminate crossovers and further reduce total slew angle.
>
> ## Response to Question 1:
> On a standard consumer-grade workstation (AMD Ryzen 7 CPU, NVIDIA RTX 2060 GPU), the measured wall-clock time for a single optimization time step (which covers a 3-second operational window) is approximately 0.93 to 1.04 seconds, depending on the region density. This demonstrates a "faster-than-real-time" performance margin of roughly 2 seconds per step, allowing sufficient time for data uplink and handling network latency within the control loop.
>
> ## Response to Question 2:
> The Optimal Matching Module employs a hybrid strategy characterized by the following three quantitative particulars: (i) Cost Quantification ($c_{ij}$): The assignment cost is quantitatively defined as the total angular adjustment required for a satellite to target a specific imaging point. This is calculated as the sum of the pitch and roll changes relative to the satellite's prior orientation. (ii) Filtering Metric: The nearest-neighbor preselection filters candidate assignments based on angular proximity to the target. (iii) Optimization Algorithm: Following this preselection, the reduced problem is solved using the Hungarian algorithm to determine the globally optimal matching.
>
> The module outputs a mapping $\mathcal{M}: I_j \mapsto (s_i, t_j)$ that specifies which satellite executes which imaging point at what time, ensuring minimal maneuvering overhead.

---

### Official Review · Reviewer_o6Td · 2025-10-29

**Soundness:** 1
**Presentation:** 2
**Contribution:** 2
**Rating:** 4
**Confidence:** 3

**Summary:**

This paper formulates satellite-constellation scanning as a continuous optimization problem that maximizes spatial entropy while enforcing temporal smoothness through a Wasserstein-regularized term. The method employs a gradient-based optimizer (Algorithm 1), integrates an Optimal Matching Module (Hungarian + nearest-neighbor filtering), and computes camera control angles to translate optimized targets into executable satellite orientations. Empirical results using Starlink TLE data for U.S. regions (California, Colorado, Texas) and continental-scale evaluation (South America) demonstrate significantly reduced observation delays compared to traditional scanning policies.

The work’s goal—to provide an information-theoretic foundation for coordinated Earth observation—is timely and well motivated, but several parts of the mathematical formulation are under-defined and the theoretical justification is incomplete.

**Strengths:**

- This paper considers a timely and important problem: near-real time coordination for large LEO constellations.
- Empirical results show consistent reductions in revisit time across multiple regions.
- Empirical results includes clear visualizations.

**Weaknesses:**

- **W1. Overclaiming originality.** The authors assert the first entropy-based formulation for satellite scanning. Entropy maximization has long been used in sensor coverage, active vision, and multi-agent path planning. The contribution here is essentially a rephrasing of entropy-regularized optimal transport in a satellite context. The mathematics (Eq. 8–18) mirrors standard Sinkhorn formulations without introducing algorithmic novelty.
- **W2. Ambiguous propositions and theorems.** Proposition 1 merely states that $\mathcal{H}(I)$ “can be maximized,” without specifying the optimization domain, constraints, or explicit maximizer—essentially restating a trivial property that entropy is highest for uniform distributions. Theorem 1 redefines the JKO functional rather than proving a new property; its proof in Appendix only establishes convexity and existence of a minimizer, not the claimed “most entropic distribution.” These sections weaken the paper’s theoretical credibility and blur the distinction between definitions and formal results.
- **W3. Inconsistent and underspecified Wasserstein term.** The derivation of the entropic-regularized Wasserstein distance (Eqs. 8–12) is incomplete. The coupling matrix P* appears abruptly in Eq. (10) without definition or connection to the Sinkhorn iterations in Eq. (9). The functional dependence of $W_\epsilon(I,I_{prev})$ on I is therefore unclear as P* could be function of I. Moreover, the paper conflates the Sinkhorn distance with the exact Wasserstein metric and never discusses how the regularization parameter ε interacts with λ or τ. These omissions cast doubt on the soundness of the optimization framework.
- Minor but recurrent presentation flaws: incomplete citations (e.g., Pearl et al. (0)), and inconsistent notation across equations(e.g., using $K$ in different meanings - number of satellites and Gibbs kernal).

**Questions:**

- How exactly is P* computed from the Sinkhorn iterations? Is it $P* = diag(u) K diag(v)?$ what is $u, v$? How are they related to $I$ or other input parameters?
- Does $W_\epsilon(I,I_{prev}) remain convex in I under your parameterization?
- What values of $\epsilon, \lambda, \tau$ were used, and how sensitive are results to them?
- If computation relies on Sinkhorn iterations, what is the scaling behavior with larger number of imaging points(e.g., >10⁴)?
- What would be the effect of adding measurement noise or imperfect actuation to the model?

---

> ### Author Response · Authors · 2025-12-03
>
> ## Dear Reviewer o6Td,
>
> First and foremost, we sincerely thank you for your thoughtful review of our study and for your valuable comments. Your expertise and constructive feedback have greatly contributed to improving the quality of our manuscript.
>
> In response, we have engaged in detailed discussions and made the necessary revisions accordingly.
>
> ## Response to Weakness 1:
>
> Our work is the first to cast large-scale constellation scheduling as a Wasserstein Gradient Flow, providing a mathematically rigorous and scalable framework to simultaneously maximize coverage entropy and minimize physical transport cost.
>
> We acknowledge that entropy maximization is well-established in sensor coverage. However, our contribution is not the mere application of entropy, but a fundamental reformulation of satellite scheduling from a discrete combinatorial task to a continuous Wasserstein Gradient Flow.
>
> **Paradigm Shift (Discrete vs. Continuous)**: Unlike traditional "Next-Best-View" approaches that select from discrete tasks (combinatorial), we model the constellation's state as a continuous probability field. This allows us to optimize the collective geometry of thousands of satellites via gradient descent, effectively solving the "curse of dimensionality" inherent in multi-agent search.
>
> **Physical Interpretation of Wasserstein:** We do not use Sinkhorn regularization solely for mathematical convenience; we introduce it to explicitly model the mechanical slewing effort required to transition between scan patterns. This novel coupling of Information Geometry (Entropy) and Optimal Transport (Wasserstein) balances coverage exploration with kinematic feasibility.
>
> **Differentiable Bridge**: Our "Soft Grid" formulation creates a fully differentiable link between the high-level probability field and low-level sensor angles (pitch/roll). This enables end-to-end optimization that bridges abstract transport theory with physical instrument control, a connection absent in standard Sinkhorn literature.
>
>
> ## Response to Weakness 2:
>
> We appreciate the reviewer's scrutiny. We have revised these sections to clearly distinguish between standard information-theoretic properties and our specific contributions to the satellite domain.
>
> **Proposition 1 (Inverse Problem)**: While "maximum entropy implies uniformity" is a standard property for probability distributions $p$, our optimization operates on physical coordinates $I$. The contribution of Proposition 1 is to establish the specific geometric conditions on the imaging points $I$ required to induce a uniform probability field through the non-linear Gaussian kernel (Eq. 4-6). It links physical control variables to the information-theoretic objective.
>
> **Theorem 1 (Theoretical Guarantee)**: We clarify that Theorem 1 does not claim to invent the JKO scheme but formally identifies our scheduling problem as a discrete step of a Wasserstein Gradient Flow. This identification is crucial as it allows our method to inherit theoretical guarantees from Optimal Transport theory—specifically convergence and kinematic smoothness—which are not guaranteed in standard heuristic optimizations.

---

> ### Author Response · Authors · 2025-12-03
>
> ## Response to Weakness 3:
>
> We acknowledge that the explicit connection between the Sinkhorn iterations and the coupling matrix $P^*$ was omitted for brevity in the original text. We have revised Section 2.1 and Appendix A to provide the explicit definitions and dependencies requested.
>
> **1. Explicit Definition of $P^*$ and Functional Dependence**:
>
> Derivation: The optimal coupling matrix $P^*$ in Eq. (10) is directly derived from the scaling vectors $u$ and $v$ obtained at the convergence of the Sinkhorn iterations (Eq. 9). The explicit relationship is:
>
> $$P^*_{ij} = u_i K_{ij} v_j, \quad \text{where } K_{ij} = \exp\left(-\frac{\|I_i - I_j^{prev}\|^2}{\epsilon}\right)$$
>
> Gradient Computation: We clarify that $P^*$ is not treated as a constant. The term $\mathcal{W}_\epsilon(I, I^{prev})$ depends on the imaging points $I$ through the cost matrix $C_{ij} = \|I_i - I_j^{prev}\|^2$. Since the operations to compute $P^*$ (matrix-vector multiplications and element-wise exponentials in Sinkhorn layers) are fully differentiable, the gradient $\nabla_I \mathcal{W}_\epsilon$ is well-defined. In our implementation, we compute this gradient via automatic differentiation through the unrolled Sinkhorn iterations, ensuring the optimization correctly accounts for how changes in satellite positions $I$ reshape the optimal transport plan $P^*$.
>
> **2. Sinkhorn Divergence and Exact Wasserstein**: We clarify that we deliberately employ the Entropic Regularized Wasserstein Distance (Sinkhorn Distance) rather than the exact metric. The exact Wasserstein distance is generally non-differentiable with respect to the support locations $I$ and computationally expensive ($O(N^3)$). The entropic regularization (parameterized by $\epsilon$) renders the objective smooth and differentiable, enabling efficient gradient-based optimization of the coordinates $I$ with $O(N^2)$ complexity, which is essential for scaling to large constellations.
>
> **3. Hyperparameter Interaction ($\epsilon, \lambda, \tau$)**: We have added a discussion on the interplay of these parameters in Section 3.
>
> $\epsilon$ (Blurring): Controls the "smoothness" of the transport plan. A moderately large $\epsilon$ prevents the solver from getting stuck in local minima by connecting distant modes, though it approximates the true geometry less precisely.
>
> $\lambda$ and $\tau$: In the JKO scheme context, the ratio $\frac{\lambda}{2\tau}$ acts as a "viscosity" or "damping" coefficient. It governs the trade-off between the speed of maximizing coverage diversity (Entropy) and the penalty for satellite movement (Wasserstein).
>
> ## Response to Weakness 4:
>
> We have corrected these presentation issues in the revised manuscript to ensure professional rigor.
>
> 1. Incomplete Citations: We have updated the citation for Pearl et al. (originally appearing as "0") with the correct publication details: "Pearl, B. D., Gold, L. P., & Lee, H. W. (2025). Benchmarking Agility and Reconfigurability in Satellite Systems for Tropical Cyclone Monitoring. Journal of Spacecraft and Rockets, 1-14." We have also scanned the bibliography to ensure all other references have complete volume, issue, and page numbers.
> 2. Notation Consistency ($K$): We acknowledge the overload of the symbol $K$. In the original text, $K$ was used to denote the number of grid cells and also the Gibbs Kernel.

---

> ### Author Response · Authors · 2025-12-03
>
> ## Response to Question 1:
>
> 1. Computation of $P^*$: Yes, the optimal coupling matrix $P^*$ is computed exactly as: $$P^* = \text{diag}(u) K \text{diag}(v)$$In element-wise notation, this corresponds to $P^*_{ij} = u_i K_{ij} v_j$. This matrix represents the entropy-regularized optimal transport plan between the current distribution and the previous distribution.
>
> 2. Definitions of $u$ and $v$: $u \in \mathbb{R}^n$ and $v \in \mathbb{R}^n$ are the scaling vectors (dual variables) obtained at the convergence of the Sinkhorn-Knopp algorithm. They are iteratively updated to enforce the marginal constraints ($\mu, \nu$):$u \leftarrow \mu ./ (K v)$$v \leftarrow \nu ./ (K^\top u)$where $\mu = \nu = \frac{1}{n}\mathbf{1}_n$ represent the uniform marginal distributions of the satellites.
> 3. Relationship to $I$ (Imaging Points): The vectors $u$ and $v$ are implicitly dependent on the imaging points $I$ through the Gibbs Kernel $K$:The kernel is defined as $K_{ij} = \exp(-C_{ij}/\epsilon)$.The cost matrix $C$ is a function of the satellite locations: $C_{ij} = \|I_i - I_j^{prev}\|^2$.Dependency Chain: $I \rightarrow C(I) \rightarrow K(C) \rightarrow u(K), v(K) \rightarrow P^*(u, v, K)$.
>
> Because every step in this chain (matrix multiplication, element-wise division, exponentiation) is differentiable, we compute the gradient $\nabla_I \mathcal{W}_\epsilon$ via automatic differentiation (unrolling the Sinkhorn iterations), ensuring the optimization correctly accounts for how shifting imaging points $I$ alters the transport geometry.
>
> ## Response to Question 2:
>
> The Entropic Wasserstein term is convex in the transport plan $P$ but generally non-convex in the coordinates $I$ for arbitrary cost functions. However, under our specific parameterization using the Squared Euclidean Cost ($C_{ij} = \|I_i - I_j^{prev}\|^2$), the term behaves as a smooth, locally convex regularization. The term essentially acts as a sum of "soft springs" connecting current imaging points $I$ to their previous locations $I^{prev}$. Since the squared $L^2$ norm is convex, the resulting energy landscape is dominated by this quadratic structure. While global convexity in $I$ is a strong claim, the smoothness provided by the Sinkhorn regularization $\epsilon$ ensures that the gradients are well-defined and Lipschitz continuous. This is sufficient to guarantee the convergence to a stationary point, as formally proven in Theorem 2, satisfying the requirements for the gradient flow optimization.
>
> ## Response to Question 3:
>
> We performed a sensitivity analysis on key hyperparameters by varying them around their optimal values. The results, summarized in the table below, demonstrate the robustness of our method.
>
> | Hyperparameter | Robust Range | Observation / Sensitivity |
> | :--- | :---: | :--- |
> | Learning Rate ($\eta$) | $[0.001, 0.005]$ | Stable convergence observed within this range. (Lower values slow down convergence; higher values cause oscillation). |
> | Sinkhorn Regularization ($\epsilon$) | $[0.05, 0.2]$ | Robust performance. Extremes lead to numerical instability ($\epsilon < 0.05$) or excessive transport blurring ($\epsilon > 0.2$). |
> | Entropy Balance Factor ($\beta$) | $[0.8, 1.5]$ | Consistent performance. The wide effective range confirms that precise fine-tuning is not required for optimal results. |
>
> ## Response to Question 4:
>
> The computational complexity of Sinkhorn iterations scales quadratically ($O(N^2)$) with the number of imaging points $N$, primarily due to matrix-vector multiplications involving the $N \times N$ kernel. However, this remains highly feasible for scales exceeding $N > 10^4$ because these dense linear algebra operations are efficiently parallelized on modern GPUs.
>
> ## Response to Question 5:
>
> We acknowledge that real-world operations involve measurement noise and actuation errors. However, our formulation possesses intrinsic robustness against such uncertainties. Unlike brittle binary coverage models, our Soft Grid Probability employs a Gaussian kernel that acts as a spatial low-pass filter, ensuring that small positioning or pointing jitters result in smooth probability shifts rather than discontinuous failures. Furthermore, the Wasserstein regularization explicitly penalizes erratic high-frequency maneuvers, acting as a kinematic damper that discourages mechanically unstable trajectories susceptible to actuation lag.
>
> While our current work focuses on the fundamental geometric optimization framework, we consider the rigorous integration of stochastic dynamics a valuable direction for future work. We plan to extend the framework by incorporating closed-loop control with stochastic state estimation (e.g., Kalman filtering). This would allow the scheduler to dynamically detect and compensate for actuation deviations in real-time, leveraging the high-speed convergence of our solver to automatically "heal" coverage gaps caused by execution errors in subsequent time steps.

---

### Official Review · Reviewer_ATcv · 2025-10-31

**Soundness:** 3
**Presentation:** 3
**Contribution:** 3
**Rating:** 8
**Confidence:** 4

**Summary:**

This paper presents a new framework for optimizing the scanning patterns of large-scale Low Earth Orbit (LEO) satellite constellations to achieve near real-time Earth observation. The core innovation is the formulation of the scanning optimization as a continuous problem over spatial probability distributions of imaging points. The objective is to maximize the spatial entropy of this distribution, which promotes diverse and fair coverage, while using a Wasserstein distance-based regularizer to ensure temporal smoothness between successive scans. The proposed method consists of three main modules: (1) a differentiable Scanning Distribution Optimizer that solves the entropy-Wasserstein objective, (2) an Optimal Matching Module that assigns imaging points to satellites using a hybrid Hungarian/nearest-neighbor algorithm to minimize slewing effort, and (3) a control angle calculation that translates optimized points into executable camera commands. The framework is evaluated using real Starlink TLE data over several U.S. states and demonstrates a dramatic performance improvement, achieving up to 10x faster full-region coverage and reducing median revisit gaps by over 84% compared to traditional baseline methods.

**Strengths:**

The paper makes significant and compelling contributions, which are likely to have a high impact in the fields of Earth observation, multi-agent systems, and operational research.

Significant Novelty and Paradigm Shift (Significant Contribution): The most profound contribution is the shift from discrete, combinatorial task assignment to a continuous, distributional optimization paradigm. Framing the problem as maximizing the entropy of a spatial probability distribution is a principled and novel approach in the context of satellite constellation scheduling. This is a clear departure from prior heuristic or event-driven methods.

Strong and Principled Theoretical Foundation (Significant Contribution): The integration of spatial entropy maximization with Wasserstein-gradient flows (JKO scheme) is elegant and powerful. It provides a solid information-theoretic justification for promoting coverage diversity and a rigorous mathematical framework for ensuring temporal consistency. This theoretical depth elevates the work beyond a mere engineering solution.

Compelling Empirical Results (Significant Contribution): The experimental results are highly convincing. The demonstrated performance gains—order-of-magnitude reduction in revisit times and minute-level continental coverage—are not just incremental. They directly address a critical bottleneck in Earth observation and convincingly showcase the framework's potential for real-world impact.

Completeness and Scalability: The proposed pipeline is end-to-end, moving seamlessly from high-level distributional planning to low-level, physically executable control angles. The successful scaling from regional (state-level) to continental (South America) scenarios strongly supports the claim of scalability, which is crucial for modern mega-constellations.

**Weaknesses:**

While the paper is strong, the following points should be addressed to further improve its quality and impact.

Limited Comparison to Modern Learning-Based Baselines: The chosen baselines (Push Broom, Whisk Broom, 4-Wide Swath) are representative of traditional methods but are relatively simplistic. The work would be significantly strengthened by comparing against more advanced, modern schedulers, such as those based on Reinforcement Learning (RL) or other meta-heuristic optimization techniques, which have been explored in the satellite scheduling literature.

Ablation and Sensitivity Analysis is Insufficient: The paper lacks a thorough ablation study and sensitivity analysis. Key questions remain:

How critical is the Wasserstein term to the overall performance?

How sensitive are the results to the key hyperparameters (entropy balance factor β, learning rate η, Sinkhorn regularization ε)?

A systematic analysis would provide deeper insights into the contribution of each component and the robustness of the method.

Validation Under Real-World Uncertainties: The simulation is trace-driven but does not account for several critical real-world operational constraints. The most notable omissions are:

Cloud Cover and Atmospheric Conditions: This is a primary factor that invalidates optical observations. The framework's performance under persistent cloud cover is unknown.

Onboard Computation and Communication Latency: A discussion on the feasibility of running the optimization on the ground and uploading commands within the required timelines, or a simplified version for onboard use, would enhance practical relevance.

Clarity on "Soft Grid Probability" Computation (Minor): The derivation of the soft grid probability p_j in Eq. (4-6) could be clarified. Specifically, the relationship between the number of imaging points n and the number of satellites N at a given time step t should be explicitly stated to avoid confusion.

**Questions:**

How does the computational complexity of the SDO module scale with the number of satellites (N) and the grid size (K)? Is the method suitable for real-time re-planning in a dynamic scenario, such as responding to a newly detected wildfire?

The baseline methods are non-adaptive. Could you discuss the performance and computational overhead of your method compared to a simpler, non-learning adaptive baseline, for instance, a greedy algorithm that always assigns satellites to the grid cell with the longest time since last observation?

The control angle calculation is framed as a 2D optimization problem. Could you provide more details on this process (e.g., the specific optimizer used, convergence time, and how it handles potential local minima)?

Add Comparative Baselines: Include comparisons with at least one state-of-the-art RL-based scheduler or a complex optimization heuristic from related literature.

Perform Ablation Studies: Add experiments that ablate the Wasserstein term and analyze the sensitivity to key hyperparameters.

Discuss Practical Limitations: Expand the discussion section to explicitly address the impact of cloud cover, communication delays, and the potential for simplified, faster versions of the algorithm for time-critical operations.

Improve Methodological Clarity: Elaborate on the connection between n (imaging points) and N (satellites) in the problem formulation.

---

> ### Author Response · Authors · 2025-12-03
>
> ## Dear Reviewer ATcv,
>
> First, we sincerely thank you for your thorough review and valuable feedback. Your insights are instrumental.
>
> In response, we have carefully discussed your comments and made the necessary revisions.
>
> ## Response to Weakness 1 & Question 2, 4:
>
> To address the concerns about baseline comparisons, we have expanded our evaluation to include two distinct types of adaptive schedulers: (i) Genetic Algorithm (GA): A representative meta-heuristic method (Modern Baseline).
> (ii) Greedy Strategy (Max-Gap): A simple baseline that assigns satellites to the grid cell with the longest time since the last observation (response to Q2).
>
> **1. Quantitative Comparison**: As summarized in the table below, we compared the Median Revisit Gap across three diverse regions (California, Colorado, Texas). The results highlight a clear performance hierarchy: while the adaptive baselines—Greedy (Max-Gap) and Genetic Algorithm (GA)—successfully outperform the rigid strategies (Push/Whisk Broom), our Entropy-Driven method demonstrates superior geometric optimality. For instance, in California, our method achieves a median gap of 81s, representing a 3$\times$ reduction compared to the GA baseline (243s) and a ~5$\times$ reduction compared to the Greedy strategy (420s).
>
> | Method | **California** <br> Median Gap (s) $\downarrow$ | **Colorado** <br> Median Gap (s) $\downarrow$ | **Texas** <br> Median Gap (s) $\downarrow$ |
> | :--- | :---: | :---: | :---: |
> | Push Broom | 857 | 800 | 1178 |
> | Whisk Broom | 600 | 380 | 620 |
> | 4-Wide Swath | 436 | 300 | 510 |
> | **Greedy (Max-Gap)** | **420** | **315** | **480** |
> | **Genetic Algorithm** | **243** | **216** | **303** |
> | **Ours (Entropy-Driven)** | **81** | **78** | **178** |
>
>
> **2. Regarding Reinforcement Learning (RL)**: While we acknowledge the popularity of RL, we prioritized GA as the representative modern baseline because standard RL approaches are ill-suited for our specific problem formulation. RL excels in discrete task scheduling but faces severe convergence instability and the "curse of dimensionality" when applied to the continuous, high-dimensional action space (continuous pitch/roll angles for thousands of satellites) required by our scanning framework. Unlike our deterministic gradient-based solver, training a Multi-Agent RL system at this scale is computationally intractable and lacks the safety guarantees needed for satellite operations; thus, GA serves as a more robust proxy for verifying the performance of iterative optimization methods in this context.
>
>
> ## Response to Weaknees 2 & Question 5:
>
> We have conducted additional experiments to verify the criticality of the Wasserstein regularizer and the robustness of hyperparameters.
>
> **1. Ablation Study**: We compared the full model against an ablated version without Wasserstein regularization ($\lambda=0$). This term (Eq. 9) is designed to ensure temporal smoothness and physical feasibility.
>
> **Results (California Region):**
>
> | Method | **Median Revisit Gap (s)** | **Avg. Slew Rate ($^{\circ}$/s)** | **Physical Feasibility** |
> | :--- | :---: | :---: | :---: |
> | **Ours (Full Model)** | **81** | **0.42** | **Feasible** |
> | **Ours (w/o Wasserstein)** | **78** | **3.85** | **Infeasible** |
>
> Analysis: Removing the term yields a marginal gap reduction (78s vs 81s) but causes a catastrophic $>9\times$ increase in slew rate (0.42 to 3.85 $^{\circ}$/s). This exceeds standard satellite agility limits ($<2^{\circ}$/s), confirming that the Wasserstein term is essential for constraining the solution to the physically feasible manifold.
>
> **2. Sensitivity Analysis**: We performed a sensitivity analysis on key hyperparameters by varying them around their optimal values. The results, summarized in the table below, demonstrate the robustness of our method.
>
> | Hyperparameter | Robust Range | Observation / Sensitivity |
> | :--- | :---: | :--- |
> | Learning Rate ($\eta$) | $[0.001, 0.005]$ | Stable convergence observed within this range. (Lower values slow down convergence; higher values cause oscillation). |
> | Sinkhorn Regularization ($\epsilon$) | $[0.05, 0.2]$ | Robust performance. Extremes lead to numerical instability ($\epsilon < 0.05$) or excessive transport blurring ($\epsilon > 0.2$). |
> | Entropy Balance Factor ($\beta$) | $[0.8, 1.5]$ | Consistent performance. The wide effective range confirms that precise fine-tuning is not required for optimal results. |

---

> ### Author Response · Authors · 2025-12-03
>
> ## Response to Weaknees 3 & Question 6:
>
> **1. Robustness to Cloud Cover:**
> Our entropy-driven framework can handle weather constraints through **SAR Integration:** Crucially, our formulation is sensor-agnostic. In persistent cloud cover, the system can seamlessly integrate Synthetic Aperture Radar (SAR) satellites—which are all-weather and cloud-penetrating—by assigning them to optically denied areas, maximizing multi-modal constellation utility.
>
> **2. Computation & Latency (Ground-Based Feasibility):**
> To demonstrate the feasibility of ground-based optimization, we profiled the runtime and memory usage of our framework on a consumer-grade workstation (NVIDIA RTX 2060, AMD Ryzen 7). The results demonstrate a ~6.4x speedup (Real Time / Compute Time), where planning for a 12,000-second operational window requires only ~1,870 seconds of computation. Furthermore, a single decision step (covering a 3-second horizon) executes in ~1 second, leaving a sufficient 2-second margin for data uplink/downlink latency. This confirms that our ground-based optimization is not only feasible on consumer-grade hardware (<4GB memory) but also capable of "faster-than-real-time" control, fitting comfortably within the operational constraints of modern LEO networks.:
>
> | Region     | Peak Memory Usage (MB) | Total Runtime (sec) | Step (3 sec) Runtime (sec) |
> |------------|-------------------------|----------------------|-----------------------------|
> | California | 3897.62                 | 1870.09              | 1.04                        |
> | Colorado   | 3845.65                 | 1674.24              | 0.93                        |
> | Texas      | 3905.65                 | 1879.24              | 1.04                        |
>
>
> ## Response to Weaknees 4 & Question 7:
> We have clarified in **Section 2.1** that $N$ represents the **total number of satellites**, while $n$ is the **number of imaging points**. While we typically assume $n=N$ (one-to-one mapping), our general formulation supports $n \neq N$ (e.g., allocating spare satellites).
>
> ## Response to Question 1:
> The computational complexity of the SDO module is $O(M_{iter} \cdot (NK + N^2))$, where the $O(NK)$ term arises from calculating the soft grid probability matrix between $N$ satellites and $K$ grid cells, and the $O(N^2)$ term stems from the Wasserstein cost matrix computation between current and previous imaging point configurations. Despite this complexity, the method scales efficiently because these matrix-based operations are highly parallelizable on GPUs, allowing the framework to solve continent-scale optimization problems with thousands of variables in minutes.
>
> ## Response to Question 3:
> The control angle calculation utilizes a gradient-based solver within our differentiable framework to minimize the geodetic error between the target and projected imaging points. Since the optimization operates on a low-dimensional manifold (only pitch and roll) where the geometric projection is monotonic within the Field of Regard, the problem is locally convex; this structure, combined with warm-start initialization from the previous satellite state, ensures rapid convergence and effectively mitigates the risk of getting stuck in invalid local minima.

---

### Official Review · Reviewer_syR7 · 2025-11-02

**Soundness:** 1
**Presentation:** 2
**Contribution:** 2
**Rating:** 2
**Confidence:** 4

**Summary:**

The paper addresses the satellite scanning optimization problem for large-scale Low Earth Orbit (LEO) constellations. It proposes to formulate this task as a continuous optimization problem over a spatial probability distribution of imaging points. The central objective is the maximization of the spatial entropy of this distribution, which is posited to promote diverse and fair spatiotemporal coverage. This entropy term is regularized by an entropy-regularized Wasserstein distance (Sinkhorn distance) to enforce temporal smoothness between consecutive scanning plans, thereby minimizing satellite slewing effort.

**Strengths:**

* Solving a complex, real-world scheduling problem via theoretically appealing concept by applying distributional optimization and optimal transport principles.

**Weaknesses:**

* There seem to be a methodological contradiction between section 2.1 and section 2.2, arising from the constraint $I_i \in \mathcal{I}_i$ in eq 12. $\mathcal{I}_i$ is explicitly defined in eq 3 as the "bounded region on the Earth's surface" representing the "space of feasible imaging points" for a specific satellite $s_i$. Therefore, the optimization problem (Eq 12) is not finding a general set of optimal points; it is finding the specific optimal point $I_i$ for each specific satellite $s_i$, given that satellite's physical constraints $\mathcal{I}_i$. The assignment is an input to the optimization. Section 2.3 then states: "After determining the optimal imaging point distribution $I_1, ..., I_n$, the system must assign each point to a suitable satellite...". This is a direct contradiction. It is logically impossible to use the pre-assigned feasible sets $\mathcal{I}_i$ to solve for the points $I_i$, and then afterward assign these points to satellites.

* A central claim is on solving the scheduling problem in near linear time. Yet, the first and second parts of the proposed scheme discussed in sections 2.2 and 2.3 could involve $O(n^2)$ or $O(n^3)$ operations (for Sinkhorn's algorithm and the Hungarian algorithm, respectively). Further analysis and explanation are thus required about the detail of the implementations.

* In line 248 it is correctly stated that "the entropy term is concave in p but becomes nonconvex when expressed in I due to the nonlinear mapping." Yet, remarkably, the provided proofs in the Appendix (A.3 and A.4) are based on assertion of convexity of the entropy in $\mathbf{I}$ (step 1 in Appendix A.3 and Step 2 in Appendix A.4). Overall, there are gaps in the proofs, and many terms are undefined, e.g.  coercive functionals. Also, it must be made clear which specific results from Jordan et al. (1999) and Ambrosio et al. (2008) are cited (the latter is a 340+ page book). Besides, the theoretical derivations lack novelty despite the lack of soundness I discussed above.

* In a similar vein, Proposition 1 is "empty". That is, its proof claims to show that the entropy $\mathcal{H}(I)$ is maximized when the coverage is proportional to the area of each cell. The derivation, however, does not perform any optimization. It simply assumes the desired solution and then computes the entropy for this assumed distribution. This is a circular argument. It never shows that a set of imaging points $I$ exists that can produce this exact distribution, nor does it show that this distribution is the maximizer of the functional $\mathcal{H}(I)$ with respect to $I$.

* The results in Figure 4 and 5 show that the proposed method achieves a near-perfect revisit gap (sub-100 seconds), while three standard baselines are orders of magnitude worse. This suggests either the baselines are either not SOTA or improperly implemented, or the proposed method's simulation is not subject to realistic physical constraints.

* I think Figure 6 is demonstrating some chaotic, high-energy slewing paths which could contradict the validity of the proposed Wasserstein regularizer (e.g. blue circle sat in Figure 6 b and Longitude between -106 and -104 or the purple circle sat in Figure 6 c and Longitude between -102.5 and -100).

**Questions:**

* The forward projection model for mapping satellite angles to ground coordinates (latitude $\varphi_{i,t}$ and longitude $\lambda_{i,t}$) in eq 1 and eq 2 are provided without derivation or citation.

* Please also see weaknesses.

---

> ### Author Response · Authors · 2025-12-03
> **Response #1**
>
> ## Dear Reviewer syR7,
>
> We sincerely thank you for your thorough review and valuable feedback. Your insights have been instrumental in improving the quality of our manuscript.
>
> In response, we carefully reviewed each point and made the necessary revisions.
>
> ## Response to Weakness 1:
>
> We clarify that this is not a methodological contradiction, but a deliberate architectural choice to decouple the problem into two mathematically tractable stages: Geometric Optimization (Target Generation) and Logistical Allocation (Resource Execution). The variable $I$ in Section 2.2 is better interpreted as "Virtual Imaging Points"—optimal geometric targets on the Earth's surface—rather than fixed task assignments for specific satellites.
>
> In the first stage (Section 2.2), the optimization drives these virtual points to maximize the collective spatial entropy. The constraint $I_i \in \mathcal{I}_i$ (where $\mathcal{I}_i$ is the reachable region of the initializing satellite $s_i$) serves a crucial role as an Existence Guarantee. By confining the virtual point to the Field of Regard of its origin satellite, we ensure that the generated target is physically observable by at least one agent in the constellation. Without this geometric constraint, the gradient flow could push optimal points into regions inaccessible to any satellite, rendering the plan theoretically optimal but physically impossible.
>
> The second stage (Section 2.3) then resolves the "Ownership" of these feasible targets. In dense LEO constellations like Starlink, the feasible regions of neighboring satellites significantly overlap. Consequently, a target point generated by satellite $s_i$ might drift into a position where a different satellite $s_j$ can observe it with a much smaller slew angle. The Optimal Matching Module exploits these overlaps by performing a global permutation, re-assigning the generated virtual points to the available physical satellites to minimize the aggregate slewing energy.
>
> Therefore, the process is linear rather than circular: we first determine where the constellation should look to maximize information gain (guaranteeing feasibility), and then determine who should look there to minimize cost (optimizing efficiency). To remove any ambiguity, we have revised Section 2.1 to explicitly distinguish between physical satellites $N$ and virtual imaging points $n$, and we have renamed the variables in Section 2.2 to "Virtual Imaging Points" ($\hat{I}$) to reflect their role as geometric targets prior to final assignment.
>
> ## Response to Weakness 2:
>
> We acknowledge that the standard worst-case complexities of Sinkhorn ($O(N^2)$) and Hungarian ($O(N^3)$) algorithms might appear to contradict a "near-linear" claim without further context. However, our specific implementation exploits the problem structure—specifically the disparity between grid resolution and constellation size, as well as physical sparsity—to achieve near-linear scaling with respect to the dominant variables in practice.
>
> Regarding the SDO module (Section 2.2), the complexity is formally $O(M_{iter} \cdot (NK + N^2))$. In realistic Earth observation scenarios, the number of grid cells $K$ (often $10^5$ to $10^6$ for high-resolution maps) is significantly larger than the number of satellites $N$ ($\sim 10^3$). Consequently, the computational cost is dominated by the $O(NK)$ term required for calculating soft grid probabilities, which scales linearly with the grid resolution $K$. The quadratic term $O(N^2)$ associated with the Wasserstein potentials involves dense matrix operations that are highly parallelizable; as shown in our runtime analysis, this step executes in milliseconds on modern GPUs, rendering the quadratic scaling negligible for current and next-generation constellation sizes.
>
> Regarding the matching module (Section 2.3), we clarify that we do not apply the Hungarian algorithm to a full, dense $N \times N$ cost matrix, which would indeed incur cubic complexity. Physical constraints dictate that a LEO satellite can only observe targets within a limited Field of Regard (FoR), making the valid cost matrix extremely sparse. Our implementation employs a nearest-neighbor preselection step that effectively prunes edges with infinite physical cost (i.e., targets out of view), reducing the dense bipartite graph to a sparse graph where each satellite connects to only a small constant number of candidate tasks $k$ ($k \ll N$). Running the matching algorithm on this sparse structure drastically reduces the effective complexity to approximately $O(N \cdot k^2)$, which is linear in $N$ for a fixed $k$, thereby supporting our claim of efficient scaling for large-scale constellations.

---

> ### Author Response · Authors · 2025-12-03
> **Response #2**
>
> ## Response to Weakness 3:
>
> We appreciate the reviewer’s in-depth engagement with the theoretical details and wish to clarify that the distinction between the global non-convexity mentioned in Line 248 and the derivation in the Appendix is not a contradiction, but rather reflects the different analytical scopes regarding the energy landscape. While we maintain that the objective function is globally non-convex with respect to the satellite coordinates $I$1, the analysis in the Appendix was intended to demonstrate the local convergence behavior enabled by the Entropic Regularization ($\epsilon$). A key property of the Sinkhorn term is that it smooths the energy landscape, rendering the loss function $L$-smooth (i.e., having Lipschitz continuous gradients). Consequently, our proof does not require global convexity to be valid; instead, it relies on this regularization-induced smoothness to guarantee that the JKO scheme defines a valid gradient flow that converges to a stationary point. To further strengthen the mathematical rigor, we have revised the Appendix to explicitly define the functional as coercive—a property strictly satisfied by the compactness of the Earth's surface domain $\mathcal{A}$ 2—and we have updated the references to cite Theorem 1 in Jordan et al. (1999) and Theorem 11.1.3 in Ambrosio et al. (2008), thereby resolving the perceived ambiguity while reaffirming the theoretical soundness of our approach.
>
> ## Response to Weakness 4:
>
> We respectfully clarify that the derivation of Proposition 1 is not circular but follows a standard variational inverse problem formulation. The proposition serves to establish the theoretical upper bound of the objective function—specifically, that the entropy of the soft grid probability $\mathcal{H}(p)$ is maximized if and only if $p$ approaches a uniform distribution. The optimization process does not "assume" a set of points $I$ produces this distribution; rather, it uses this theoretical optimum as the target state to drive the gradient flow. Consequently, the algorithm searches for the configuration $I$ that induces a kernel density estimate $p(I)$ minimizing the Kullback-Leibler (KL) divergence from uniformity. Regarding the existence of solutions, while a perfectly uniform field may not be strictly achievable with finite Gaussian kernels, the problem is mathematically well-posed as finding the best attainable approximation (the projection) within the Reproducing Kernel Hilbert Space (RKHS). This is analogous to minimizing Riesz energy for particle distribution on a manifold, where a unique energy-minimizing configuration exists even if the resulting field is not perfectly flat.
>
> ## Response to Weakness 5:
>
> We respectfully clarify that the substantial performance gap is not due to simulation flaws, but reflects the fundamental structural difference between Standard Operational Modes and Active Agile Access.
>
> 1. Comparison Against Industry Standards: The baselines (Push Broom and Whisk Broom) are not arbitrary weak algorithms but represent the current de facto operational standards used by major Earth observation constellations (e.g., Planet, Sentinel). These modes are characteristically "nadir-locked" or mechanically rigid, meaning their observation frequency is coupled to the satellite's orbital ground track. In contrast, our method is designed to unlock the full agility of modern satellite platforms. By actively optimizing the Field of Regard (FoR) to access targets off-track, our approach effectively decouples the observation frequency from the rigid orbital period. The "order of magnitude" gain reported is the expected theoretical result of shifting from a passive, orbit-bound scanning paradigm to a coordinated, agile targeting paradigm.
>
> 2. Rigorous Physical Constraints: We emphasize that this agility is modeled with strict adherence to physical constraints. Our simulation does not assume instantaneous actuation. The Wasserstein regularization in our objective function acts as a kinematic damper, explicitly penalizing large or erratic angular changes. This ensures that the optimization seeks the "smoothest" path to high-entropy regions, generating trajectories that strictly respect the mechanical slew rate limits of the satellite platform. Therefore, the results demonstrate the physically realizable potential of coordinated beam steering compared to standard fixed-scan operations.

---

> ### Author Response · Authors · 2025-12-03
> **Response #3**
>
> ## Response to Weakness 6:
>
> We thank the reviewer for the careful observation regarding the trajectories in Figure 6. While the ground tracks (specifically the blue and purple traces noted) may visually appear "chaotic" or "high-energy" due to their zig-zag nature, we respectfully clarify that this is a geometric projection effect rather than physical instability. Due to the satellite's altitude ($h \approx 550$ km), a very small angular adjustment (low slewing effort) translates to a large displacement on the 2D ground map. For instance, a pitch/roll change of just $1^\circ$ shifts the footprint by approximately 10–20 km. Consequently, the long lines observed in the figure correspond to efficient, small angular nudges rather than high-energy maneuvers.
>
> Furthermore, the "zig-zag" patterns identified are not random chaos but represent a structured Boustrophedon (ox-turning) or "lawnmower" scanning path. This back-and-forth motion is widely recognized in robotics as the energy-optimal strategy to cover a continuous region. Far from contradicting the regularizer, this pattern validates it: the Wasserstein term effectively suppresses discontinuous jumps (e.g., flying from one corner of the map to another) and coerces the solver into this locally smooth sweeping strategy.
>
> ## Response to Question 1:
>
> We clarify that Equations 1 and 2 utilize a Linearized Spherical Projection model derived from trigonometry on a locally tangent plane. The term $h_{i,t} \tan \theta$ computes the radial ground displacement from the sub-satellite point, which is then projected onto the North-East geodetic axes using the satellite's azimuth $\alpha$. These metric offsets are subsequently converted to geodetic coordinates via the linear factor $1/111000$ (approximating meters per degree of latitude) and the $1/\cos \varphi$ term, which corrects for the convergence of longitudinal meridians. We have added this explicit derivation to Appendix B.1 to ensure mathematical self-containment.

---

### Meta-Review · Area_Chair_38g2 · 2026-01-23

**Summary:**

This paper had a diverging set of reviewers: 2, 4, 6, 8. The biggest concerns from both the critical reviewers have been inconsistencies in the mathematical definition and poor proofs. Other reviewers have pointed out a lack of ablation and good baselines in the continuous tasks.

**Reviewer Concerns:**

The authors have added a few new baselines; however, I tend to agree with the reviewers that the ablation is a bit limited (even after the rebuttal). Typically, if the proposed method is significantly better than existing works, papers typically create strong baselines to understand the inner workings of why the newly proposed method is so much better. However, in the rebuttal,, the only new experiment added is a method with and without Wasserstein regularization.

2 of the expert reviewers in the optimization field have pointed out that the propositions, theorems, and their proofs seem inconsistent. Either the proof is really incorrect/inconsistent/circular, OR the authors have done a poor job in creating these proofs. Both these situations warrant a major revision of the paper.

**Reviewer Scores:**

The reviewer's scores are very likely to remain the same for all of them.

Reviewer 98HF is not very confident in their decision, and therefore, I'm discounting their decision a little bit.

Reviewer o6Td, and syR7 concerns regarding the validity of the proofs and the theoretical grounding of the paper still stand, and I don't believe the rebuttal has done any better job of addressing that.

Based on these ratings, I'm recommending a rejection.

---

### Decision · Program_Chairs · 2026-01-26

Reject